# Examining the Eastern European extreme summer temperatures of 2023 from a long-term perspective: the role of natural variability vs. anthropogenic factors

Monica Ionita[1,2*], Petru Vaideanu[2,3], Bogdan Antonescu[4,2,3], Catalin Roibu[2], Qiyun Ma[1] and Viorica Nagavciuc[1,2]

[1]Alfred Wegener Institute Helmholtz Centre for Polar and Marine Research, Bremerhaven, 27570, Germany
[2]Faculty of Forestry, Ștefan cel Mare University, Suceava, Romania
[3]Faculty of Physics, University of Bucharest, Măgurele, 077125, Romania
[4]National Institute for Earth Physics, 12 Călugăreni, 077125 Măgurele, Romania.

*Correspondence to*: Monica Ionita (Monica.Ionita@awi.de)

**Abstract.** Amidst unprecedented rising global temperatures, this study investigates the historical context of heatwave (HW) events in Eastern Europe. The record-breaking 2023 summer, featuring a HW lasting for 19 days in the south-eastern part of Romania, extending up to Ukraine, necessitates a deeper understanding of past extreme events. Utilizing statistical methods on long-term station data spanning from 1885 to 2023, we aim to detect and analyze historical HWs, particularly focusing on events predating 1960. This extended timeframe allows for a more comprehensive assessment of noteworthy extremes compared to recent decades. We used both a percentile-based threshold and a fixed absolute temperature threshold to identify HW events. Our analysis identifies two critical periods with increased HW frequency and intensity: 1920–1965 and 1980–2023, respectively, highlighting the most extreme events in August 1946, August 1952, July 2012, June 2019, and August 2023. Furthermore, reanalysis data shows that historical HWs, similar to the 2023 event, were associated with large-scale European heat extremes linked to high-pressure systems and they were accompanied by extreme drought, thus leading to compound extreme events. We find that while a clear trend emerges towards more frequent HWs from the 1980s onward, the analysis also uncovers substantial HW activity on daily timescales throughout the 1885-1960 period. Moreover, we highlight the intertwined impacts of climate change and multidecadal internal variability on HW patterns, with evidence suggesting that both contribute to the increasing frequency and intensity of these extreme events. Attribution analysis reveals that the extreme summer temperatures observed in 2023, would not have been possible in the absence of anthropogenic climate change. Regardless of future warming levels, such temperatures will occur every year by the end of the century. Our research highlights the value of extending the historical record for a more nuanced understanding of HW behavior and suggests that extreme heat events, comparable to those experienced in recent decades, have occurred throughout the analyzed period.

## 1 Introduction

Global warming poses a significant threat, as indicated by the projected increase in the magnitude and frequency of HW events across Europe and globally (Fischer and Schär, 2010; IPCC, 2021a; Perkins and Alexander, 2013; Russo et al., 2019; Manning et al., 2019; Doshi et al., 2023). These events have devastating consequences, exemplified by the designation of HWs as the leading cause of weather-related fatalities in the US by the National Weather Service (Robinson, 2001; IPCC, 2021a). Europe has also seen a marked increase in both the frequency and intensity of these events in the last decades, with severe socio-economic consequences. For example, ~ 70,000 lives were lost in the 2003 European HW (Robine et al., 2008; García-Herrera et al., 2010), while the 2022 summer HW in Europe and the UK caused an estimated 15,000 and 3,200 deaths, respectively (Ballester et al., 2023). The impact of heat extremes is not equally distributed, with race, class, and gender disparities playing a significant role in vulnerability (Benz and Burney, 2021; Chakraborty et al., 2019). These disparities can be attributed to factors such as the urban heat island effect, varying levels of tree cover and green spaces across socioeconomic

groups, and the density of the built environment. While a single hot day may not significantly increase mortality, consecutive days of extreme heat, particularly with high nighttime temperatures, can lead to a substantial rise (Perkins, 2015; Barriopedro et al., 2023; Ma et al., 2024).

Eastern Europe in particular, a region historically characterized by moderate temperatures, has witnessed a concerning rise in the frequency and magnitude of HWs in recent decades (Ionita et al., 2021; Nagavciuc et al., 2022b; Croitoru et al., 2016; Croitoru and Piticar, 2013). These extreme weather events pose a significant threat to human health, ecosystems, and infrastructure. Furthermore, in Eastern Europe, there are many urban areas with limited green spaces, which suffer disproportionately during HWs, demanding immediate attention and comprehensive mitigation strategies. Several factors contribute to the increasing prevalence of HWs in Eastern Europe. Climate change intensified by human activities such as the emission of greenhouse gases, alongside natural variability, play a pivotal role in the occurrence, magnitude and frequency of these events (Luo et al., 2023). Rising global temperatures create conducive conditions for the development and persistence of high-pressure systems, leading to stagnant air masses and amplifying HWs and droughts (Han et al., 2022; Bakke et al., 2023; Ma and Franzke, 2021). Additionally, regional factors like changes in the atmospheric and oceanic circulation patterns coupled with land-use practices further exacerbate the issue (Vautard et al., 2023; Vaideanu et al., 2020). The consequences of HWs in Eastern Europe are far-reaching and multifaceted, demanding immediate attention and comprehensive mitigation strategies. The most immediate impact is on mortality and human health, with increased risks of heatstroke, dehydration, and cardiovascular complications, particularly among vulnerable populations like the elderly and young children (European Environment Agency, 2022; Vicedo-Cabrera et al., 2021). HWs also disrupt agricultural production, leading to crop failures and impacting food security (Malik et al., 2022). Furthermore, they exacerbate drought conditions, straining water resources and increasing the risk of wildfires (Zscheischler et al., 2018; Brando et al., 2019; Ionita and Nagavciuc, 2020; Laaha et al., 2017). Given these pressing challenges, this study delves into the multifaceted phenomenon of HWs in Eastern Europe, exploring their causes and impact.

Common definitions of HWs rely on exceeding fixed temperature thresholds or deviations from normal values, such as daily mean or maximum temperatures (Cowan et al., 2014; Barriopedro et al., 2023; Alexander, 2016). Understanding these definitions is important for accurately monitoring HWs and implementing effective response strategies. Here, we use a historical approach to analyze HWs occurrences since the 19[th] century in the eastern part of Europe, with a particular emphasis on Romania. Examining past events offers valuable insights into the decadal variability of extreme events (e.g., HW frequency and magnitude, cold spells, storms) as documented by previous studies (Beckett and Sanderson, 2022; Hawkins et al., 2023; Lorrey et al., 2022; White et al., 2023). Analyzing past extreme events expands the temporal sample for future studies, fostering a deeper understanding of the underlying mechanisms driving such phenomena. This expanded knowledge base allows also for improved preparedness for similar events in the future, even those not readily captured by current climate models (Van Oldenborgh et al., 2022). Additionally, historical analysis transcends the limitations of solely studying present-day climate, enabling researchers to contextualize modern events like the recent European summer HWs (e.g., 2003, 2015, 2019, 2022) within a broader historical framework (Yule et al., 2023; Hegerl et al., 2019). Therefore, this research leverages the valuable information gained from past events to enrich our understanding of HW activity, ultimately contributing to improved preparedness and contextualization of contemporary extreme heat events over Europe.

This study focuses on analyzing HWs in Romania from the 19[th] century to the present, specifically from May through September. Individual historical HWs are compared to contemporary events, such as the one from 2023 , to explore potential shifts in their characteristics (e.g., frequency, duration and magnitude). Long-term daily maximum temperature data served as the foundation for identifying HWs in the target period. Here, we make use of both of the already available data, as well as newly digitalized data from old meteorological year books (Ionita and Nagavciuc, 2024). A regionally specific definition of a

HW is employed to optimize detection accuracy. Further, reanalysis data is used to analyze the spatial extent and synoptic conditions associated with historical HWs. This integrated approach, provides a comprehensive assessment of the efficacy of long-term regional datasets in capturing large-scale HW events. Section 2 details the observational data and methodology employed in our study. Section 3 presents the main results. The discussion and the concluding remarks are presented in section 4.

**2 Data and methods**

HWs can be defined either by utilizing a threshold-based methodology (Perkins, 2015) or by using the exceedance of a fixed absolute value (e.g., daily maximum temperature ≥30 °C) (Robinson, 2001). The threshold-based method identifies HWs as periods where the daily maximum temperature (TX) exceeds a specific percentile threshold for that calendar day, taking into account the regional temperature variations. The fixed temperature threshold method defines a summer as any day when the maximum temperature surpasses a fixed value (e.g., 30°C). In our study, we used both the threshold-based method, specifically employing the 90[th] percentile of daily maximum temperatures within a 15-day window centered around each calendar day (Perkins and Alexander, 2013), and the fixed absolute method for the number of days with Tx ≥ 30°C (i.e., summer days). For the threshold-based method, we have tested different HW durations (3-6 consecutive days) and selected a 5-day period for our analysis. HW duration thresholds vary globally to reflect regional climates: for example, in Canada (2+ days), Hungary and France (3+ days), China and Ukraine (5+ days). Our choice of a 5-day threshold is suitable for Romania's location in Eastern Europe and our emphasis on extreme events (Nagavciuc et al., 2022b). This aligns with the recommendations of the Expert Team on Climate Change Detection and Indices (ETCCDI) and allows us to focus on disastrous, extreme HW events. Our baseline for calculating the 90[th] percentile was the period of 1971-2000. We also used the fixed treshold method to count the number of summer days with the daily maximum temperature ≥30°C(TX30). We used the daily maximum temperature records at 31 meteorological stations over the period 1961 – 2023 (Figure 1), and a dataset of longer-time meteorological records covering the period 1885 – 2023. A detailed description of the long-term meteorological stations and the period covered by daily TX records is given in Table 1.

The daily maximum temperature for the meteorological stations (see Table 1) are taken from the European Climate Assessment & Dataset (ECA) (Klein Tank et al., 2002). The long-term meteorological records are a combination of the time series extracted from ECAD and newly digitalized daily meteorological records extracted from the yearly reports of the Romanian meteorological service (Hepites S., 1899; Ionita and Nagavciuc, 2024), which have been homogenized using the *climatolR* package (Guijarro et al., 2023) in order to ensure data consistency and reliability. Next to the station-based daily records from Romania (Table 1) we also use the gridded daily maximum temperature from the E-OBS dataset (Cornes et al., 2018). Additionally, the hourly data for the Universal Thermal Climate Index (UTCI), which in an indicator for the outdoor thermal comfort levels, has been computed based on the ERA5 dataset (Di Napoli et al., 2021). To analyze the large-scale atmospheric circulation pattern associated with the occurrence of HWs, we use the daily and monthly geopotential height anomalies at 500mb level (Z500) and the zonal and meridional wind at the same level. For the Z500 and the corresponding wind patterns associated with particular HW events (see Section 3) we made use of the ERA5 dataset (Hersbach et al., 2020), while for the long-term analysis (i.e., 1885 – 2023) we made use of the NOAA/CIRES/DOE 20th Century Reanalysis (V3) (NCEPV3) (Slivinski et al., 2019). The ERA5 dataset covers the period 1940 – 2023 and has a spatial resolution of 0.25° x 0.25°, while the NCEPV3 covers the period 1885 – 2015 and has a spatial resolution of 1° x 1°. In order to analyze the drought conditions associated with the occurrence of extreme HWs we have used the Standardized Precipitation Evapotranspiration

Index (SPEI) for an accumulation period of 1-month (SPEI1) (Vicente-Serrano et al., 2010). The precipitation and evapotranspiration dataset needed to compute SPEI, have been extracted from the E-OBS dataset (Cornes et al., 2018).

The large-scale anomaly patterns associated with long-term changes in the frequency of summer days are analyzed through composite map analysis (von Storch and Zwiers, 1999). The composite maps are computed for the years when a certain index is higher (lower) than +0.75 (−0.75) standard deviation. The difference between these averages is the composite map, where the differences are the anomalies. A simple t-test determines the anomalies' statistical significance.

## 3 Results

### 3.1 Variability, change and drivers of summer days (TX30)

This sub-section investigates the changes in extreme summer temperatures at the national level within Romania. A linear trend analysis was conducted on the number of summer days exceeding 30°C (TX≥30°C) at 9 meteorological stations possessing long-term data (details in Table 1). The analysis considered both the entire data period (varying per station) and the common timeframe of 1961-2023 (presented in Figure 2). Figure 2 shows a statistically significant upward trend in the number of summer days at all stations, particularly during the 1961-2023 period. Notably, Bucuresti recorded the most substantial increase, with summer days rising by approximately 8.8 days per decade, followed by Calarasi, with a 7.6-day increase per decade. Looking at the broader period, starting from 1896, Tg. Magurele witnessed the most significant increase (~3.1 days per decade), while Arad came in second with an increase of ~2.9 days per decade over the same period. The identified trends consistently point towards an increase in summer days across Romania over the past 140 years. Analyzing the temporal evolution from a long-term perspective reveals also multi-decadal variability. Two notable periods stand out: 1920-1970 and 1980 onwards, both characterized by a rise in summer day frequency.

The 30-years probability density function (PDF) for some of the long-term meteorological stations shows a very clear pattern: the period 1991 – 2023 was the hottest one (e.g., in terms of frequency of summer days) over the observational record, while the period 1961 – 1990 was the least warm one (e.g., in terms of the frequency of summer days). The difference in the number of summer days, at Bucuresti station, between (1991 – 2023) and (1961 – 1990) is approximately 26 days (Figure 3a), which is statistically significant (p<0.001, based on a two-sample Wilcoxon rank test). At Calarasi there is an increase of approximately 25 days (the highest one) in the frequency of summer days over the period 1991 – 2023 relative to the period 1961 – 1990. Overall, the period 1991 – 2023 is the time frame with the highest frequency of summer days at all stations, the period 1961 – 1990 is the time frame with the smallest number of summer days, while the periods 1901 – 1930 and 1931 – 1960 have a similar distribution in terms of summer days. The difference in the number of summer days between the periods 1991 – 2023 and 1961 – 1990 is statistically significant at all analyzed stations (Figure 3).

Across all analyzed stations, a positive trend is evident for the summer days, with the strongest increase observed in the south-eastern part of the Romania. However, considerable multidecadal variability is evident in the number of days with TX>30°C. The dominant mode of multidecadal variability in the climate system is the Atlantic Multidecadal Oscillation (AMO, (Kerr, 2000)). In its positive phase, the AMO is characterized by warmer temperature anomalies over the North Atlantic (Enfield et al., 2001; Drinkwater et al., 2014; Knight et al., 2006), significantly affecting the Northern Hemisphere's climate (Dima and Lohmann, 2007), including precipitation (Vaideanu et al., 2018), land temperature (Knight et al., 2005; Ionita et al., 2013) or sea-ice (Vaideanu et al., 2023). Therefore, variations in the sign and magnitude of the AMO could be responsible for modulating the summer days over Romania. To test this hypothesis, we applied the EOF technique for the time series of TX30 at 9 stations over a common period (i.e., 1923 – 2023) and we have looked at the first EOF and its corresponding

temporal evolution (i.e., PC1) (Figure 4a). We choose the first EOF (EOF1), as it captures 93% of the total variance and extracts the common signal from each station. This mode captures an in-phase variability for all analyzed stations and the corresponding eigenvalues for each station are shown in Table 2. This monopolar structure (positive values at all stations) suggests that mainly the large-scale atmospheric and oceanic circulation influences the multidecadal variability of the summer temperature over Romania. Its temporal evolution (PC1) smoothed with a 21-year running mean filter, resembles the evolution of the summer (MJJAS) AMO index (Figure 4a) and the correlation coefficient between the smoothed PC1 and the smoothed summer AMO index is 0.94 (99 % significance level). A period with an increased frequency of summer days (1925–1965), was followed by a prolonged period with a reduced number of summer days (1965 – 1990). The last ~30 years of the analyzed interval are also characterized by a higher frequency of summer days. High (low) frequency of summer days occurs during the warm (cold) phase of the AMO index. This builds on our previous work (Ionita et al., 2013) where we have shown that the multidecadal variability of the summer mean temperature over Romania is mainly driven by the phase of the AMO.

In order to understand how the sea surface temperatures (SSTs) might influence the frequency of summer days in Romania, composite maps of summer (MJJAS) SST were created for the years with high (>1 SD) vs low (<-1 SD) PC1 values. The difference map (High – Low, Figure 4b) shows a quasi-monopolar North Atlantic signal, with positive SST anomalies across the basin during high PC1 years. This aligns with the PC1-AMO correlation and with well-known AMO-related SST patterns (Mestas-Nuñez and Enfield, 1999; Enfield et al., 2001; Knight et al., 2006; Drinkwater et al., 2014). Considering the structure of the composite map, we argue that during the warm (cold) phase of the AMO, Romania experiences an increase (decrease) in the frequency of summer extreme temperatures, which is in agreement with the evolution of the summer PC1 time series, presented in Figure 4a. Additionally, we used Z500 anomalies and wind vectors for years marked by increased summer temperatures, identifying a pattern of positive Z500 anomalies over Europe. The composite map of the Z500 anomalies and the corresponding wind vectors for the years with increased frequency in summer days is characterized by a large center of positive (negative) Z500 anomalies over the central and eastern parts of Europe (central North Atlantic Ocean) (Figure 4c). The spatial structure of the Z500 anomalies, with the positive center over the eastern part of Europe, favors the advection of dry and warm air from the east or southeastern part of Europe, reduced precipitation, and an increased frequency of high temperatures over the regions situated under the influence of the anticyclonic circulation. This can lead to the development or strengthening of a HW as they cause clear skies, calm winds, and allow the air to warm significantly, often through subsidence (Ma and Franzke, 2021; Ionita et al., 2022). This observation is consistent with previous research (Gao et al., 2019), which have shown that during the positive phase of AMO a center with Z500 anomalies over the central and eastern part of Europe co-exists with a hot spot for extremely high temperatures over the same regions. Thus, we argue that the warm phase of AMO influences the frequency of HWs over the eastern part of Europe, including Romania, via the modulation of the large-scale atmospheric and oceanic patterns. In summary, a significant upward trend in summer days exceeding 30°C in Romania is observed, with the most notable increases in the southeastern regions and during the period 1961-2023. The AMO also emerges as an important driver, influencing these temperature trends through its impact on sea surface temperatures and large-scale atmospheric patterns and underscoring its pivotal role in modulating Romania's climate extremes.

## 3.2 Variability and change in the occurrence of long-term summer HWs

The strongest HWs, detected using the threshold-based method, are visualized using bubble plots, where the cumulative intensity (i.e., the sum of all the temperature anomalies throughout the duration of the HW) determines the rank. Figure 5 shows the cumulative intensity (the sum of daily maximum temperature anomalies throughout a HWs) plotted against its duration (represented by circle size) for ten Romanian meteorological stations over the period 1885-2023. Some stations,

like Arad (Figure 5a) and Bucuresti (Figure 5c), demonstrate a gradual increase in heat intensity over time. Others, like Calarasi (Figure 5e) and Cluj Napoca (Figure 5f), exhibit a mixed pattern with high and low heat intensity years. Interannual variability is evident across all stations (e.g., Bucuresti (Figure 5c) showing significantly higher cumulative heat intensity in 2019 and 2023 compared to surrounding years). Figure 5 suggests a positive correlation between HW duration and cumulative intensity; longer HWs tend to have higher cumulative intensity. However, there is considerable variability within the data, including short HWs with high cumulative intensity and long HWs with lower intensity. A striking feature of Figure 5 is the clustering of HWs over two periods: 1920-1965 and 1980-2023. This indicates that alongside interannual variability, Eastern European HW frequency exhibits a multidecadal component, which can be influenced by the state of SSTs in the North Atlantic Basin as shown in Section 3.1. As in the case of summer days, the clustering of HWs occurs over the same period when AMO was in its positive phase (i.e., 1920 -1960 and 1990 – present), while the periods with a reduce number of HWs occurred over the same time when AMO was in its negative phase (i.e., mainly between 1960 until 1990) (Figure 4a – red line). Overall, the temporal evolution of cumulative intensity as a function of HW duration over the past 140 years further highlights a denser concentration of high-intensity HWs within the past three decades (Figure 5).

The 2023 HW emerged as the strongest event in terms of duration and cumulative intensity at 10 out of 31 stations analyzed, marking it as an exceptional event in recent decades (Table 3). In Bucuresti, this event was exceptional in both magnitude (cumulative intensity of 165.26°C) and duration (19 days). The second-strongest HW was recorded in August 1946 (Figure 4 and Table 4), lasting 15 days with cumulative intensities of 125.81°C (Buzau), 128.73°C (Tg. Jiu), and 115.55°C (Calarasi). Additional long-lasting HWs of high magnitude occurred in August 1952, June/July 2012, and June 2019 (Table 3 and Table 4). Section 3.4 will provide a detailed analysis of these specific HW events.

### 3.3 Then vs. now

Based on the two HW prone periods identified in Figure 5, we have computed the temporal changes in the HWs metrics over two periods: 1920 – 1965 and 1980 – 2023, respectively. When looking at the distribution of the duration (Figure 6a) and number of HWs (Figure 6b) over these two periods, we notice a significant increases in both metrics over the period 1980 – 2023 with respect to the period 1920 – 1965. The strongest increase (~212% / 194%) is found for the HW duration/number at Bucuresti station, followed by Arad (186% / 180%) and Tg. Magurele (186% / 203%). All analyzed stations indicate an increase both in the duration and the number of HWs between the two periods, an increase which is statistically significant (p<0.001, based on a two-sample Wilcoxon rank test). The increase in the overall duration of HWs varies between 141% (i.e., Cluj Napoca) to 212% (i.e., Bucuresti) and for the number of HWs from 136% (i.e., Cluj Napoca) to 194% (i.e., Bucuresti). The stations situated outside the Carpathian Arch (i.e., Bucuresti, Buzau, Calarasi, Tg. Jiu, Tg. Magurele and Dr. Tr. Severin) show a higher increase in the duration and number of HWs compared to the station situated inside the Carpathian Arch (e.g., Arad, Baia Mare, Cluj Napoca and Vf. Omu), emphasizing the strong influence the Carpathian Mountains have on the climate of Romania. In terms of cumulative intensity (Figure 7), the same pattern can be observed: the strongest difference, between the two analyzed periods, for the cumulative intensity is found at Bucuresti Filaret station (215%), followed by Tr. Magurele (184%), while the smallest difference can be observed at Tg. Jiu (142%) and Cluj Napoca (143%).

If we consider only the period 1961 – 2023, when more meteorological stations are available and perform the same analysis by splitting the data into two parts, namely 1961 – 1990 and 1990 – 2023, the situation becomes even more critical (Figure S1 and S2). At Bucuresti station an increase in the HW duration of 461% has been found, together with an increase of 400% in the number of HWs and an increase of 492% in the cumulative intensity. All analyzed stations (see Table 1) indicate an increase of at least 250% in the duration, number and cumulative intensity over the period 1991 – 2023 compared to the period 1961 – 1990, but the most affected ones are the stations situated in the south and eastern part of the country. Here

we show only the 10 stations as in the previous sub-section, but the analysis has been performed at all 31 meteorological stations. This exceptional increase in the HW metrics (i.e., duration, number and cumulative intensity) when considering only the period 1961 – 2023 indicates how misleading it is to quantify the real extent and change in the HWs metrics, if one considers short periods. Although the difference is also statistically significant and reaches values of up to 200%, it is valuable and indicated to perform HWs-related statistics on long-term time series.

**3.4 Large-scale drivers of extreme HWs**

In this section, an in-depth analysis of the most extreme HWs (e.g., 1946, 1952, 2012, 2019 and 2023, respectively) observed for particular months, in respect to their large-scale drivers is presented. We specifically examine case studies in August 1946, August 1952, June/July 2012, June 2019 and August 2023. An extended HW impacted Romania in August 1946, characterized by record-breaking temperatures and lasting up to 14 days in specific regions (particularly the south). The HW was first observed on the 10th of August in western Romania (Figure S4a), spreading eastward across the country over subsequent days. Peak temperatures were observed on August 13th (Table 4, Figures 8a and 4a). The cumulative HW intensity (CHI) reached values as high as 130°C in southeastern Romania and even 160°C over Ukraine (Figure 8b). Regionally, Tg. Jiu exhibited the highest CI (128.73°C), followed by Buzau (125.81°C) and Calarasi (115.55°C). This event coincided with approximately 200 hours exceeding the heat stress index UTCI of 32°C, signifying "very strong heat stress" (Figure S5a) across most of the country (excluding mountainous areas). Summer 1946 stands out also as one of Romania's hottest and driest on record (Nagavciuc et al., 2022a). Drought severity ranged from moderate to extreme nationwide (Figure 8d). A significant co-occurrence of intense heat and drought was observed over Romania and Ukraine (Figures 8b, 8c and 8d). The anomalies in large-scale atmospheric circulation during the HW suggest the transport of warm air masses from Russia by northeasterly airflow, leading to the extreme temperatures (daily maximum anomalies up to ~12°C, Figure S4a) and coinciding with drought conditions (Figure 8d). Next to the extreme HW event, the summer of 1946 and the following months have been characterized by an excessive famine. The lingering effects of World War II, corroborated with extreme heat and dryness severely compromised agricultural capacity. Major grain-producing regions, encompassing Romania, Ukraine, Moldavia, the lower and middle Volga basins, Rostov Oblast, and the central black earth zone, experienced a crippling drought. This resulted in a precipitous decline in crop yields (Wheatcroft, 2012). For example, the grain harvest in the Soviet Union yielded only 39.6 million tons, which represented a significant decline compared to the previous year's harvest of 47.3 million tons and a dramatic reduction from the 95.5 million tons harvested in 1940, the last full year before World War II (Ganson, 2009).

The year 1952 goes down in history for an astonishingly warm summer. On the16th of August, it was 41°C in Timișoara (western part of the country) and 39.4°C in Satu Mare (north-western part of the country), which marks the August records for these stations. At Arad meteorological station (Figure 9a) the maximum daily temperature was also recorded on the 16th of August (i.e., 40.4°C) and the HW lasted for 15 days (i.e., 3.08 – 17.08.1952). The maximum intensity of this event, on the 16th of August 1952, reached anomalies of up to 12°C over the north-western part of Romania and western Ukraine (Figure S4b). This HW was mainly focused on the eastern part of Europe (Figure S6), with the most affected regions over Ukraine, Romania, Hungary and Poland (Figure S6). The cumulative intensity reached a value of up to 120°C over the western part of Romania (Figure 9b) corroborated with up to 200 hours of UTCI>32°C (Figure 9c), indicating very strong heat stress over these regions. On the peak day of the HW (i.e., the 16th of August), most of the western part of Romania was under "very strong heat stress" (Figure S5b), while the eastern part was under "strong heat stress" conditions. As in the case of the August 1946 event, the August 1952 was accompanied by extreme drought conditions, with a center over the eastern part of Europe (Figure 9d), which led to the occurrence of a compound event with significant stress on the agriculture and society (Topor,

1963). The occurrence of the August 1952 compound hot and dry event was the consequence of an anomalous high-pressure center extending from Russia until the easter part of Europe, which led to the advection (horizontal transport) of hot and dry air from Eurasia towards the eastern part of Europe (Figure 9e).

Summer 2012 was characterized by a series of HWs. The first one lasted for 7 days (i.e., June 17th – June 23rd), followed by the longest HW for this summer, which lasted for 16 days between June 30th and July 15th (Figure 10a). Another 2 HW events occurred in August, lasting for 6 and 7 days, respectively, while the last HW was recorded in September and lasted for 10 days (Figure 10a). The longest HW which started at the end of June and finished in the middle of July, triggering exceptionally high temperatures, with daily maximum temperature anomalies reaching up to 12°C in central Europe (Figures S4c and S7). The extremely hot conditions initially impacted the western part of Europe before also encompassing the eastern part of Europe, including Romania, with a particular emphasis on plains and plateaus. The cumulative intensity of the HW reached values up to 160°C (Figure 10b), with Romania being in the center of the HW. Also, the heat stress was extremely high throughout the duration of the HW, with up to 200 hours of very strong heat stress and higher at country level, except for the Carpathian Mountains (Figure 10c). On the day of the HW peak (i.e., July 15th) the southern part of the county reached a daily maximum temperature anomaly >12°C (Figure S4c) and was under "very strong heat stress" (Figure S5c). The extreme heat was accompanied by extreme drought (SPEI1 <-2) over the southern part of the county and severe drought (SPEI1 >-1) over the northern part (Figure 10d). In general, the spatial extent of the HW (e.g., the eastern part of Europe) is mirrored by dry conditions over the same regions (Figure 10d). The dominant atmospheric circulation pattern during this period featured a northeasterly flow (Figure 10e), which facilitated the advection (horizontal transport) of warm air masses originating from Russia towards the eastern and central part of Europe. At a national scale, this large-scale atmospheric pattern established anomalously high temperatures and a surge in the number of hot days (temperatures exceeding 35°C), particularly in the southern and eastern regions (Nagavciuc et al., 2022b). The persistence of a high-pressure system positioned over Russia extending up to Romania was the primary drivers of the excessive temperatures and dryness.

June 2019 saw one of the most extensive (in spatial coverage) and prolonged HW ever recorded over Europe at that time (Sánchez-Benítez et al., 2022; Xu et al., 2020; Nagavciuc et al., 2022b). The central and eastern parts of Europe experienced exceptional warmth through the duration of the HW, with daily temperature anomalies exceeding 12°C for most of June (Figure S8). The June 2019 HW event started on the 8th of June (Figure 11a) and lasted for 19 days until the 27th of June. Romania was affected by this event especially over the central and northern parts of the country, with a cumulative intensity of up to 160°C in the northern part of the country and up to 180°C over Poland (Figure 11b). At country level, the meteorological stations with the highest cumulative intensity were: Bistrita (139.87°C), Cluj Napoca (141.18°C), Constanta (137.28°C) and Tulcea (147.34°C) (see Table 3). The peak of the HW was on the 14th of June, when the daily maximum temperature anomalies exceeded more than 10°C over large areas in Europe and up to 12°C over the northern-western part of Romania (Figures S4d and S8). Over the regions situated at low latitudes (i.e., out of the range of the Carpathian Mountains), the cumulative heat stress factor reached values up to 150 hours over the duration of the HW (Figure 11c) and on the day of the peak of the HW the same regions were affected by "strong heat stress" (Figure S5d). Compared to previously described HW events (i.e., 1946, 1952 and 2012) the drought conditions in June 2019 prevailed over most of Europe, except for the British Isles, north-western part of Fennoscandia and Turkey were wet conditions prevailed (Figure 11d). The areas covered by severe to moderate drought in June 2019 are similar to the areas affected by the long-lasting HW event (Figure 11b and 11d). The large-scale atmospheric circulation anomalies throughout the duration of the HW event were characterized by positive geopotential height anomalies over the central and eastern parts of Europe, flanked by negative Z500 anomalies over

the central North Atlantic and north-western Russia (Figure 11e). The spatial structure of the Z500 anomalies resembles a classical omega blocking pattern, which promotes the advection of warm and dry Saharan air towards the southern and eastern part of Europe (Figure 11e).

Summer 2023 was the warmest on record globally by a large margin, with an average temperature of 16.77°C (i.e., 0.66°C above global average relative to the climatological period 1971 - 2000) and a series of HWs were experienced in multiple regions of the Northern Hemisphere, including southern Europe, the southern United States, and Japan (https://climate.copernicus.eu/summer-2023-hottest-record). Over Romania there were 3 separate HW events that occurred between June until the end of September. The first events started on the July 10[th] and lasted for 13 days (Figure 12a). The second event was much longer, lasting for 19 days between August 16[th] until September 3[rd] (Figure 12a), while the third HW occurred between September 20th and lasted until the October 1[st]. The longest HW (i.e., 19 days) was characterized by unprecedented intensity and duration at 10 stations, out of the 31 analyzed in this study (Table 3). At Bucuresti Filaret, the August HW reached a cumulative intensity of 165.26°C, which is the highest CHI since 1885 (Table 4). The peak of the HW was on the August 28th, when the daily maximum temperature anomalies exceeded more than 12°C over large areas in the eastern part of Europe (Figure S4d and S9). The cumulative intensity over the eastern part of Europe reached values up to 160°C over Ukraine and southern part of Romania (Figure 12b), and the cumulative heat stress factor reached values up to 200h over the duration of the HW over these regions, except the areas situated at higher altitudes (Figure 12c). On the day of the peak of the HW the south-eastern part of Romania and western Ukraine were affected by "very strong heat stress" (Figure S5e). The areas covered by severe to moderate drought in August 2023 (Figure 12d) are similar to the areas affected by the long-lasting HW event (Figure 12b). The large-scale atmospheric circulation anomalies throughout the duration of the HW event are similar to the previous cases, namely a center of positive Z500 anomalies over the central and eastern part of Europe and a center of negative G500 anomalies over the western Russia, which leads to the advection of hot and dry air from the south (Figure 12e).

### 3.5 Attribution of the 2023 extreme summer temperatures

The extended summer temperature anomalies (i.e., May – September) indicate that year 2023 was on average with +3.1°C warmer at Baia Mare station (i.e., relatively to the climatological period 1971 – 2000), with +4.1°C warmer at Bucuresti, with +2.8°C warmer at Calarasi station, with +2.9°C warmer at Cluj Napoca station, with +2.6°C warmer at Tg. Jiu station and with +3.8°C warmer at Dr. Tr. Severin station. To assess the contribution of anthropogenic climate change to the extreme summer temperatures of 2023, we applied the methodology outlined by (Rantanen et al., 2024). This approach allows meteorologists and researchers to quantify the climate change signal on mean temperatures at the country or station level, across different timescales: monthly, seasonal, and annual. It integrates observational monthly mean temperature data with simulations from the Coupled Model Intercomparison Project Phase 6 (CMIP6) climate models, and for this study, we focus on two Shared Socioeconomic Pathways (SSPs): SSP2-4.5, and SSP5-8.5, respectively. A comprehensive description of the tool and the CMIP6 models employed is provided by (Rantanen et al., 2024). We utilized this method due to its straightforward applicability, particularly for station-based observational data, and because of the temperature record at some of our analyzed stations (i.e., Baia Mare, Bucuresti, Calarasi, Cluj Napoca, Tg. Jiu and Dr. Tr. Severin) spans a sufficiently long period (i.e., from 125 years up to 167 years) to support this type of analysis. In the current study, we focused on the whole extended summer (i.e., the summer mean temperature averaged over the months May-June-July-August-September). This tool utilizes two widely used metrics in attribution studies: the probability ratio (PR) and the change in intensity (ΔI). The PR measures the

increase in the likelihood of an event due to climate change, while ΔI represents the extent to which climate change has altered the event's intensity. To calculate ΔI, the observed temperature's percentile within the current climate distribution is first identified, and then the corresponding temperature for the same percentile is located within the climate distribution of the year 1900.

The attribution analysis suggests that the likelihood of experiencing an extended summer (i.e., May- September) as warm as the one in 2023 under the climate conditions of 1900 varies between 0.0% at Baia Mare and Cluj Napoca (i.e., such a warm summer would have been impossible to happen in the climate around the year 1900) up to 0.3% at Tg. Jiu station (*Table 5 and Table 6*). In today's climate, the probability of the observed extended summer 2023 mean temperature is higher than in the past, with a likelihood between 0.1 % at Bucuresti Filaret up to 16.3% at Dr. tr. Severin station (*Table 5 and Table 6*). An extended summer that is at least as warm was that of 2023, will occur on average once every 6 to 82 years, depending on the station analyzed. For example, at Bucuresti and Baia Mare station, the temperatures recorded in 2023 would have been impossible with the effect of climate change, while at Tg. Jiu and Dr. Tr. Severin stations the probability of such a warm extended summer has increased by a factor of 45.3 and 9.6, respectively.

Based on CMIP6 projections under the SSP2-45 scenario, by 2100, an extended summer as warm as that of 2023 is expected to occur roughly once every year at all analyzed stations (Table 5), with the exception of Bucuresti station, where such a warm summer is expected to occur once every 3 years. The same holds true in the case of the SSP5-85 scenario (Table 6), including Bucuresti station. Under the SSP5-85 scenario, a summer as warm as the one in 2023 is expected to occur every year, at all analyzed stations. Thus, this implies that by the end of the 21$^{st}$ century, summers as warm as the one in 2023 will be the new normal (Figure 13 and Figure 14). This highlights the significant shift in temperature extremes anticipated by the century's end, as depicted in Figure 13 and Figure 14. Additionally, our analysis of intensity changes (ΔI) indicates that the extended summer of 2023 was approximately 1.6°C warmer than it would have been without anthropogenic climate change at Baia Mare, Bucuresti, Calarasi and Cluj and ~1.7°C warmer at Tg. Jiu and Dr. Tr. Severin.

**4 Discussion and conclusions**

Eastern Europe has experienced a succession of extreme HWs in recent years, notably in June/July 2012, June 2019 and August 2023 (Nagavciuc et al., 2022b; Russo et al., 2019, 2015; Lhotka and Kyselý, 2022; Ionita et al., 2021). These HWs have surpassed historical records in intensity and duration, causing severe impacts on human health, ecosystems, and infrastructure and have been driven mainly by persistent large-scale patterns (e.g., increased frequency of atmospheric blocking) (Bakke et al., 2023; Kautz et al., 2021; Rousi et al., 2023; Lau and Nath, 2012; Yang et al., 2019; Chan et al., 2022). In line with these studies here we show that in all analyzed HW events (i.e., August 1946, August 1952, June/July 2012, June 2019 and August 2023) the area affected by the HW was under the influence of a persistent high-pressure system which led to the advection of hot and dry air either form Eurasia or from Sahara. This spatial structure in the Z500 intensifies incoming solar radiation, leading to extremely high temperature anomalies under the high-pressure center. These findings are in agreement with previous studies which have shown that hot and dry summers are usually accompanied by an increase in the frequency and persistence of atmospheric blocking (Bakke et al., 2023; Ionita et al., 2021; Kautz et al., 2021; Schubert et al., 2014; Ma and Franzke, 2021). Scientific analysis points to a combination of large-scale atmospheric patterns and the amplifying effects of climate change as the primary drivers of these events (Gao et al., 2019; Luo et al., 2023; Ma et al., 2024).

From a large-scale atmospheric/oceanic point of view, the AMO can favor, in its positive phase, an increase in the frequency of HWs over the eastern part of Europe (Gao et al., 2019). However, the remarkable increase in the HW frequency over the past 30 years extends beyond what can be attributed solely to AMO's influence. The rate of increase in the metrics of

the HWs, varying between ~200% up to 400%, for all analyzed stations, especially since 1961 onwards, points to greenhouse gases induced warming as a significant contributor. This aligns with the findings of a recent study (Luo et al., 2023) indicating that ~43% of the recent increase in the HW frequency over the central and eastern part of Europe was driven by the positive phase of the AMO, while ~57% is related to greenhouse gases induced warming. The observed trend strongly suggests the influence of a combined effect from anthropogenic climate change and natural modes of internal variability

We provide compelling evidence of the escalating intensity and frequency of HWs patterns in Romania over the last century, with clear indications of climate change. The clustering of HWs also aligns with AMO phases, with more frequent and intense HWs occurring during its positive phase, underscoring the significant role of internal variability in modulating Eastern European HWs. We also show whether any past occurrences rival the intensity of modern events. Analysis of observational data from 1885 to 2023 revealed an increase in both the number of days exceeding the 90$^{th}$ percentile temperature threshold and the overall frequency of HWs events. We found an increase of at least 200% (depending on the station) in the HW metrics for the recent events (i.e., 1980 – 2023) compared to previous periods characterized by a high frequency of HWs (i.e., 1920 – 1965). This aligns with the observed warming trend in monthly average temperatures observed both globally as well as at European level (IPCC, 2021b). Despite the observed upward trend in the summer days and HW metrics over Romania, in this study we also found that certain historical HWs were comparable to recent events in terms of mean and maximum temperatures (i.e., August 1946 and August 1952 events). These past events offer valuable case studies and point in the direction that such intense HWs are not exclusive to the modern era, highlighting also the significant influence of natural variability, such as AMO. Over Romania, the most extreme HW on record occurred in August 2023, with a maximum daily temperature >35°C for 13 consecutive days at Bucuresti Filaret meteorological station (i.e., 17.08.2023 – 29.08.2023). The second most intense August HW, in terms of duration and intensity, occurred in 1946. Reanalysis data reveal that historical HWs often coincide with widespread heat and dry conditions across Europe under the influence of extensive high-pressure anomalies. Moreover, by employing a tool for attribution-based analysis, we have shown that the extended summer temperature recorded in 2023 would have been impossible without the contribution of anthropogenic warming and such warm summers will become the new normal by the end of the 21$^{st}$ century.

The urgency for both reducing greenhouse gasses emissions and implementing robust adaptation measures across Romania is irrefutable. Continued research and collaboration between scientists, policymakers, and stakeholders are essential to address this growing threat and build a more resilient future for the region. We also emphasize the need for further studies (e.g., by employing long-term meteorological time series) in other European regions to gain a broader understanding of escalating HW risks and strengthen pan-European preparedness efforts. Recognizing the gravity of this issue, various stakeholders must collaborate to develop effective mitigation and adaptation strategies. Implementing climate change mitigation measures to reduce greenhouse gasses emissions is crucial for long-term solutions. This paper provides a comprehensive understanding of HWs in eastern Europe by examining their variability, trends and causes in a long-term perspective.

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

*Competing interests*. The authors declare that they have no conflict of interest.

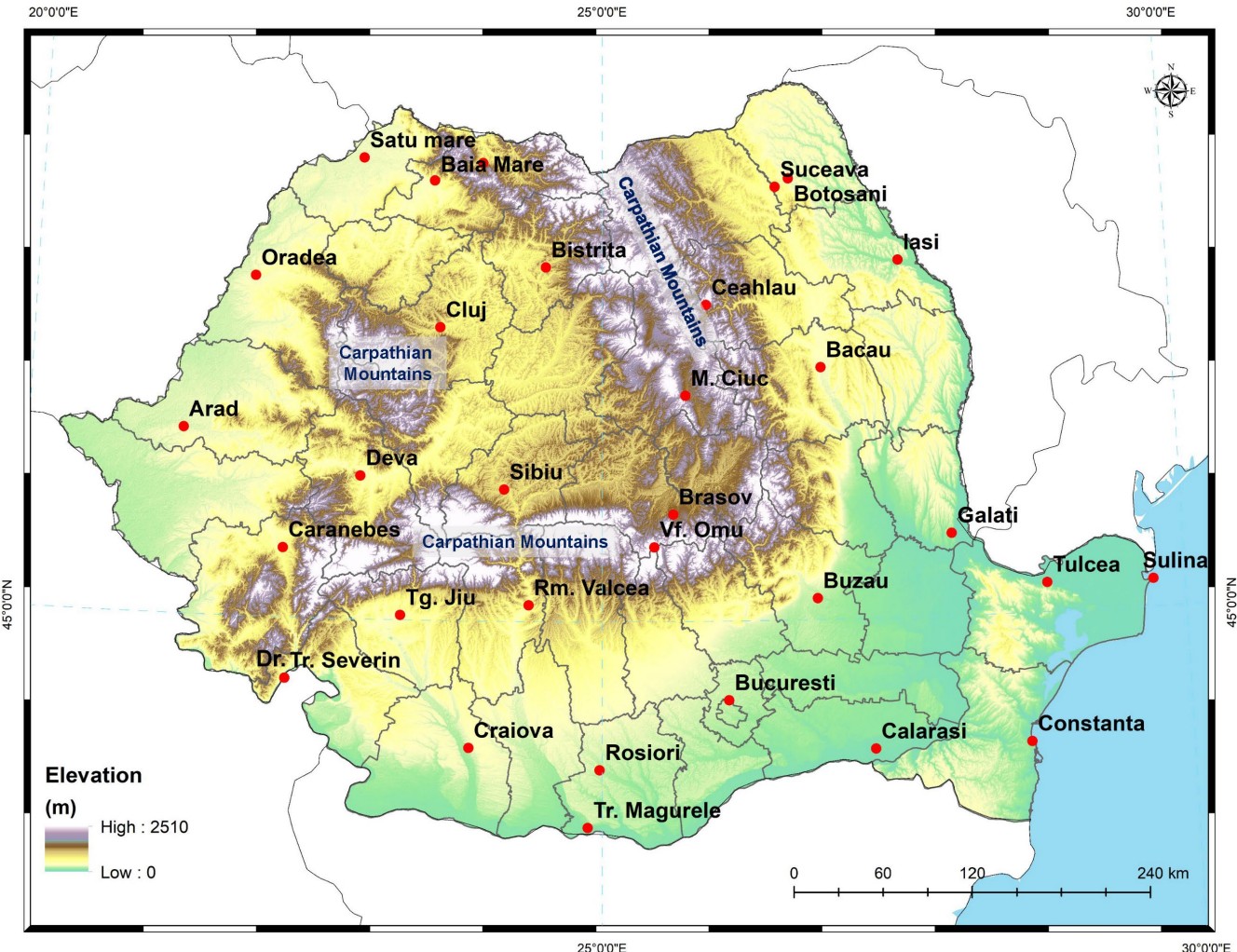

**Figure 1**. Location of the meteorological stations used in this study and the location of the Carpathian Mountains

**Number of days with daily Tx≥30°C**

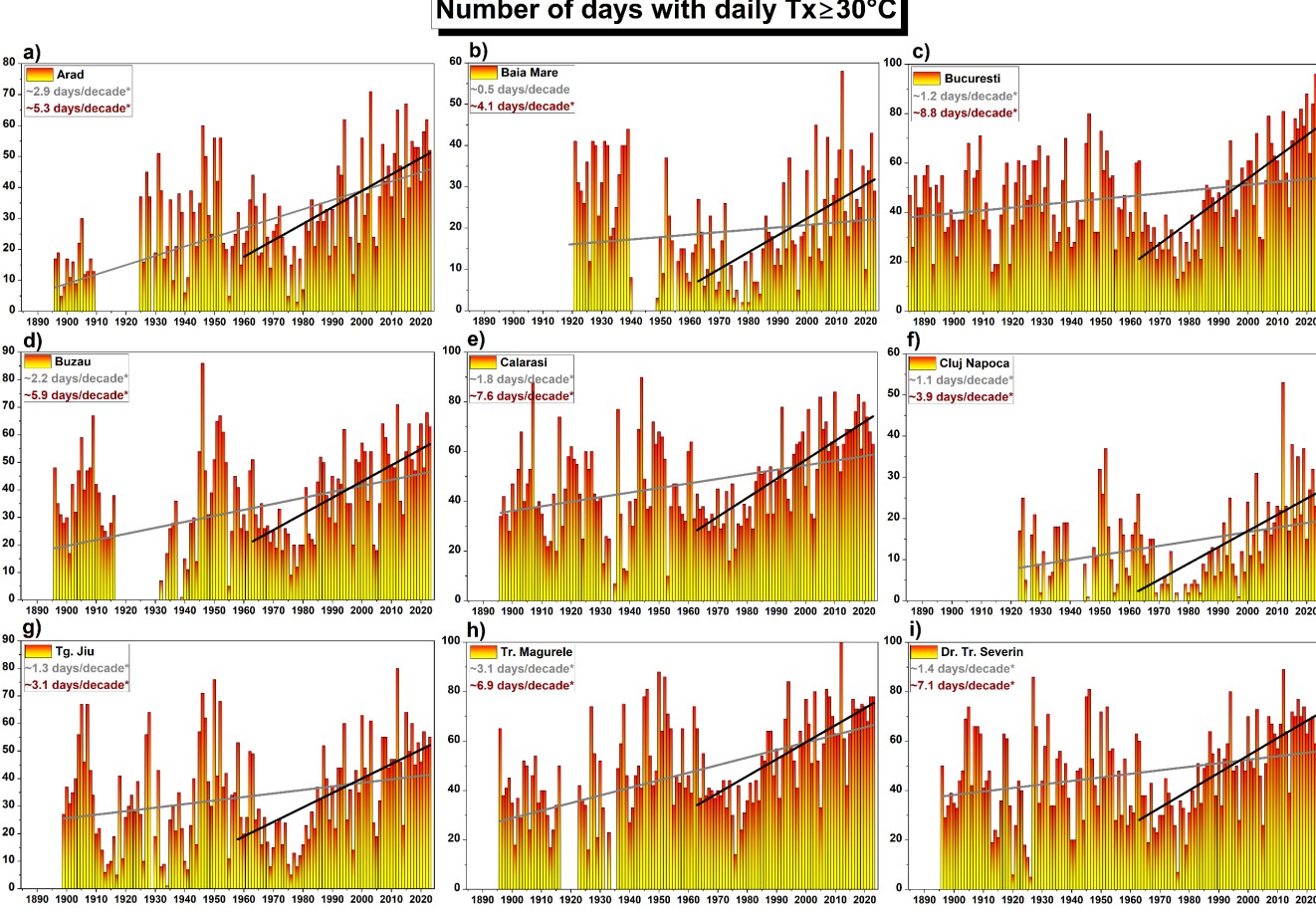

*Figure 2*. The total number of days with daily maximum temperature above 30°C over the period 1885 – 2023 and the linear trend over the period 1885 – 2023 (grey line) and over the period 1961 – 2023 (red line). a) Arad; b) Baia Mare; c) Bucuresti; d) Buzau; e) Calarasi; f) Cluj; g) Tg. Jiu; h) Tr. Magurele and i) Dr. Tr. Severin. Analyzed time frame: May – September. Yellow colors indicate smaller values, while red colors indicate higher values.

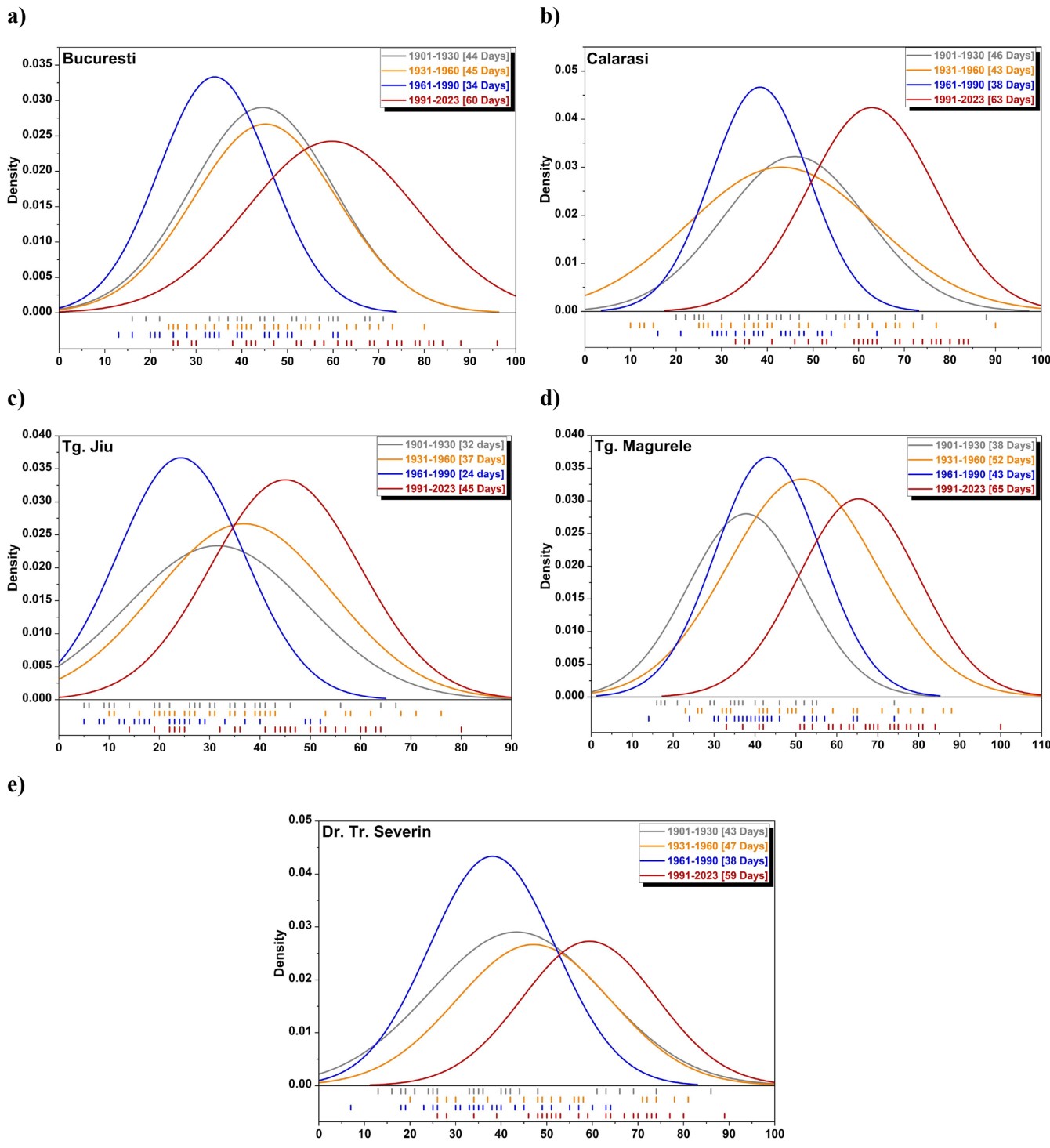

***Figure 3***. The probability distribution function of the number of days with TX ≥ 30°C for different periods (i.e., 1901 – 1930, 1931 – 1960, 1961- 1990 and 1991 – 2023, respectively); a) Bucuresti, b) Calarasi, c) Tg. Jiu, d) Tr. Magurele and e) Dr. Tr. Severin. The numbers in brackets represent the mean values. Analyzed time frame: May – September.

**a)**

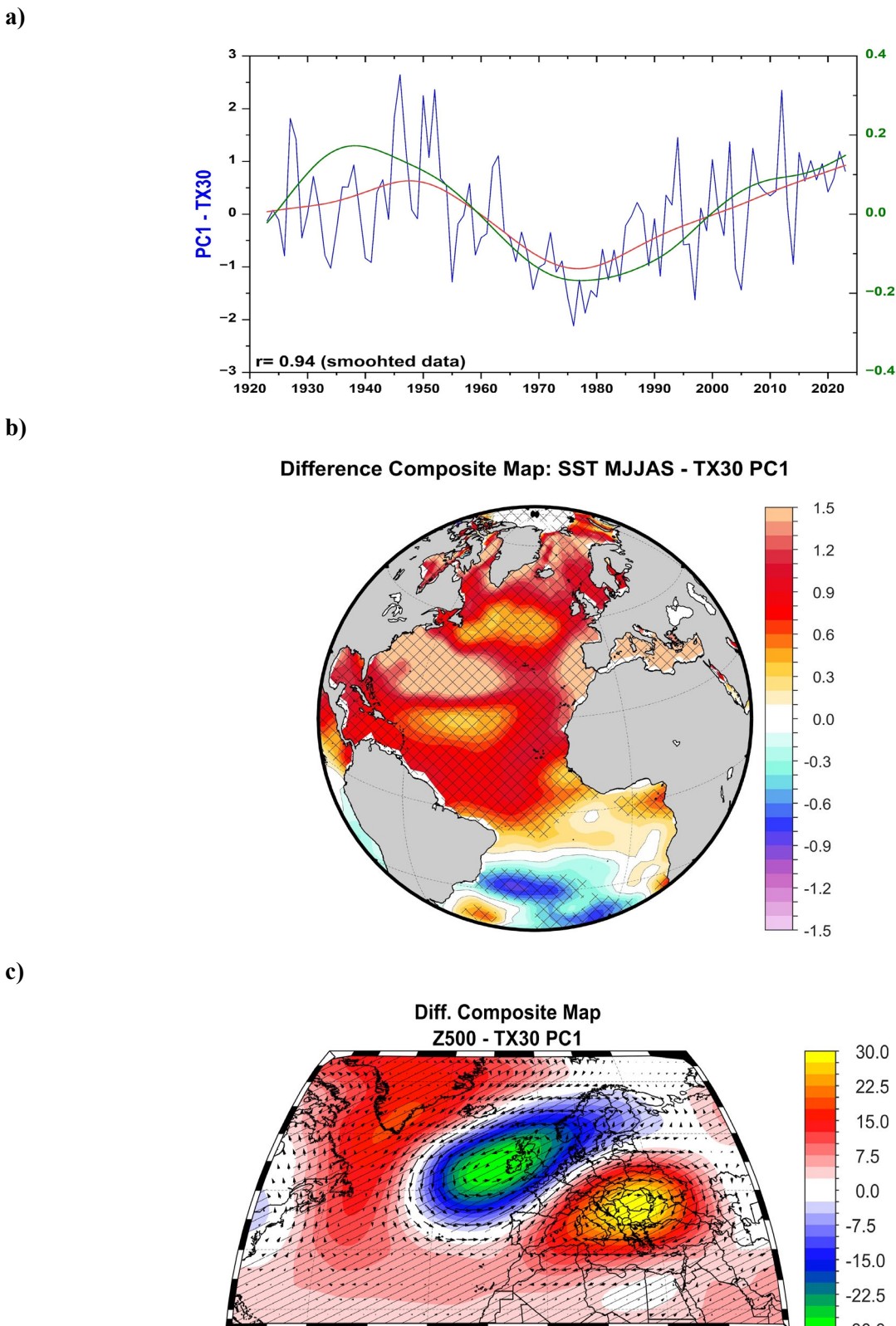

**b)**

**c)**

***Figure 4***. a) The time series of the raw PC1 (blue line), the smoothed PC1 (green line) and the smoothed time series of summer (MJJAS) AMO index (red line). The time series were smoothed using the Whittaker-Henderson approach; b) The composite map (High – Low) between PC1 and the normalized SST. Before the composite map analysis, a running mean of 5 years has been applied to PC1 and the SST field, to focus on the decadal to multidecadal variability; and c) The composite map (High – Low) between PC1 and the corresponding 500mb geopotential height (Z500) and winds vectors. The hatched areas indicate anomalies significant at the 95% significance level based on a *two-tailed t-test*. Units: Z500 [m].

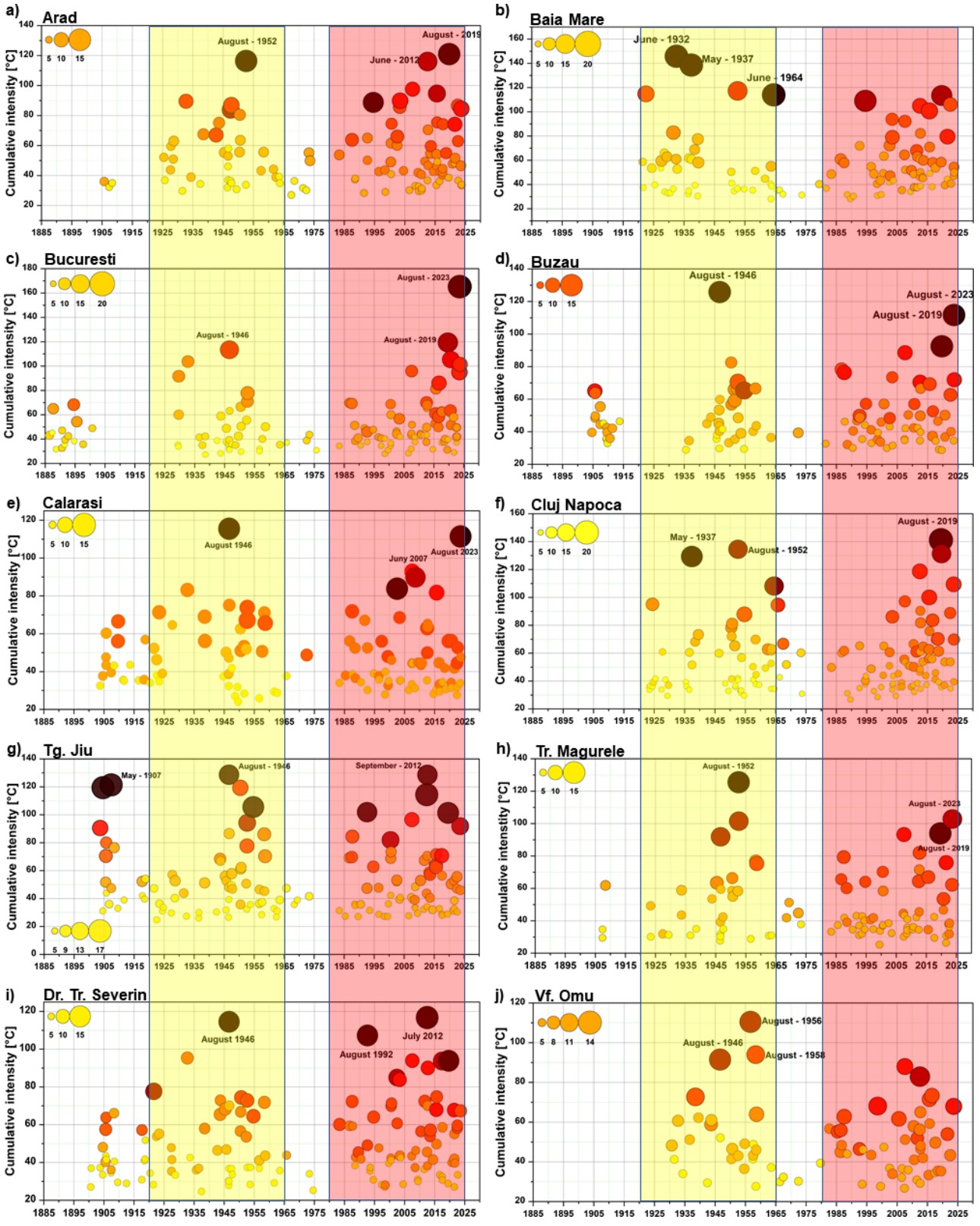

***Figure 5***. The cumulative intensity [°C] as a function of the duration of each HW (size of the circles). a) Arad; b) Baia Mare; c) Bucuresti; d) Buzau; e) Calarasi; f) Cluj; g) Tg. Jiu; h) Tr. Magurele; i) Dr. Tr. Severin and j) Vf. Omu. Analyzed time frame: May – September.

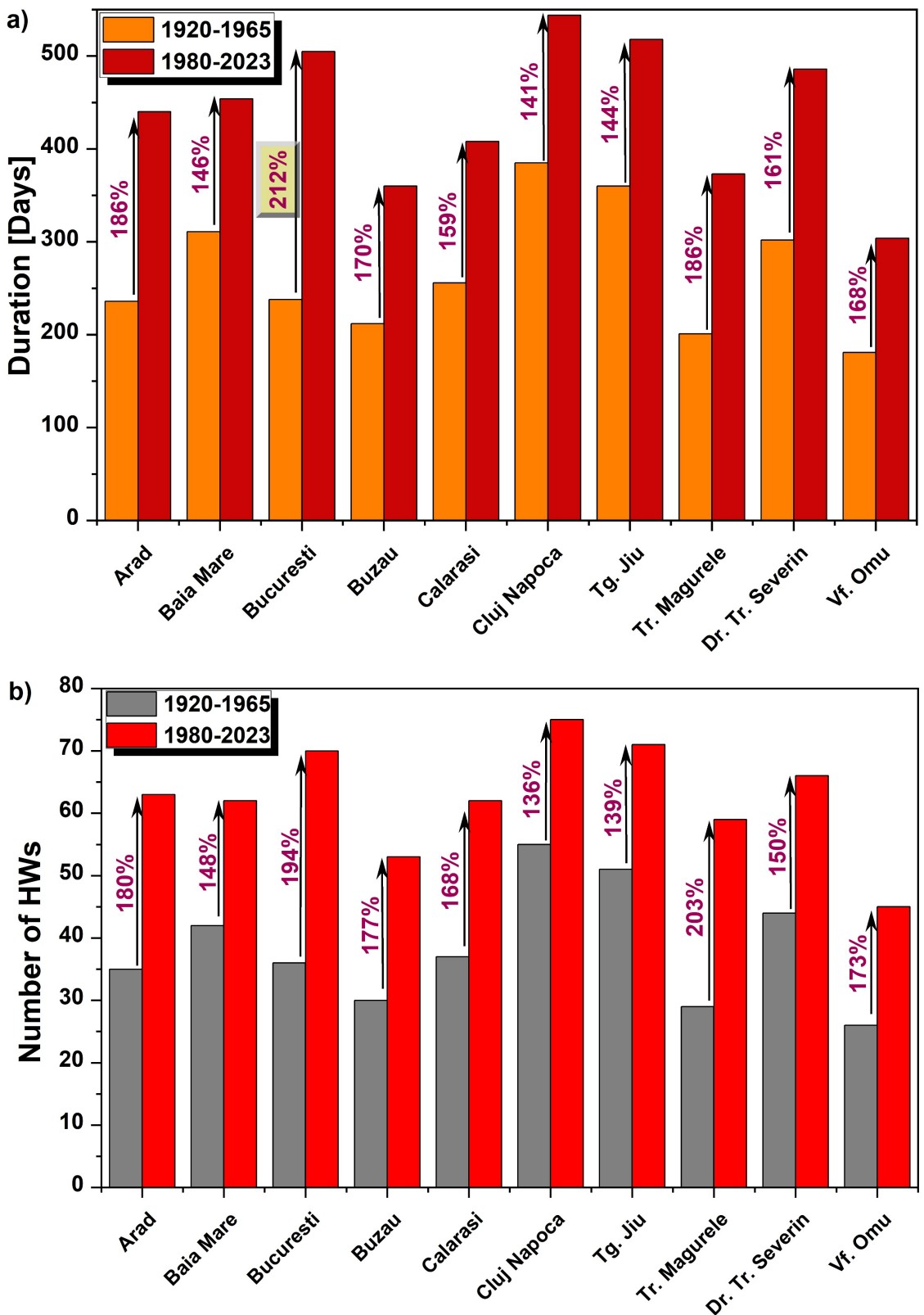

**Figure 6.** a) Distribution of the duration (i.e., sum of all days affected by a HW) over two periods, namely 1920 – 1965 (orange bars) and 1980 – 2023 (red bars), respectively and b) Distribution of the number of HWs (i.e., sum of all HWs) over two periods, namely 1920 – 1965 (gray bars) and 1980 – 2023 (red bars), respectively. The black arrows in a) and b) indicated the rate of change (as %) between the two analyzed periods.

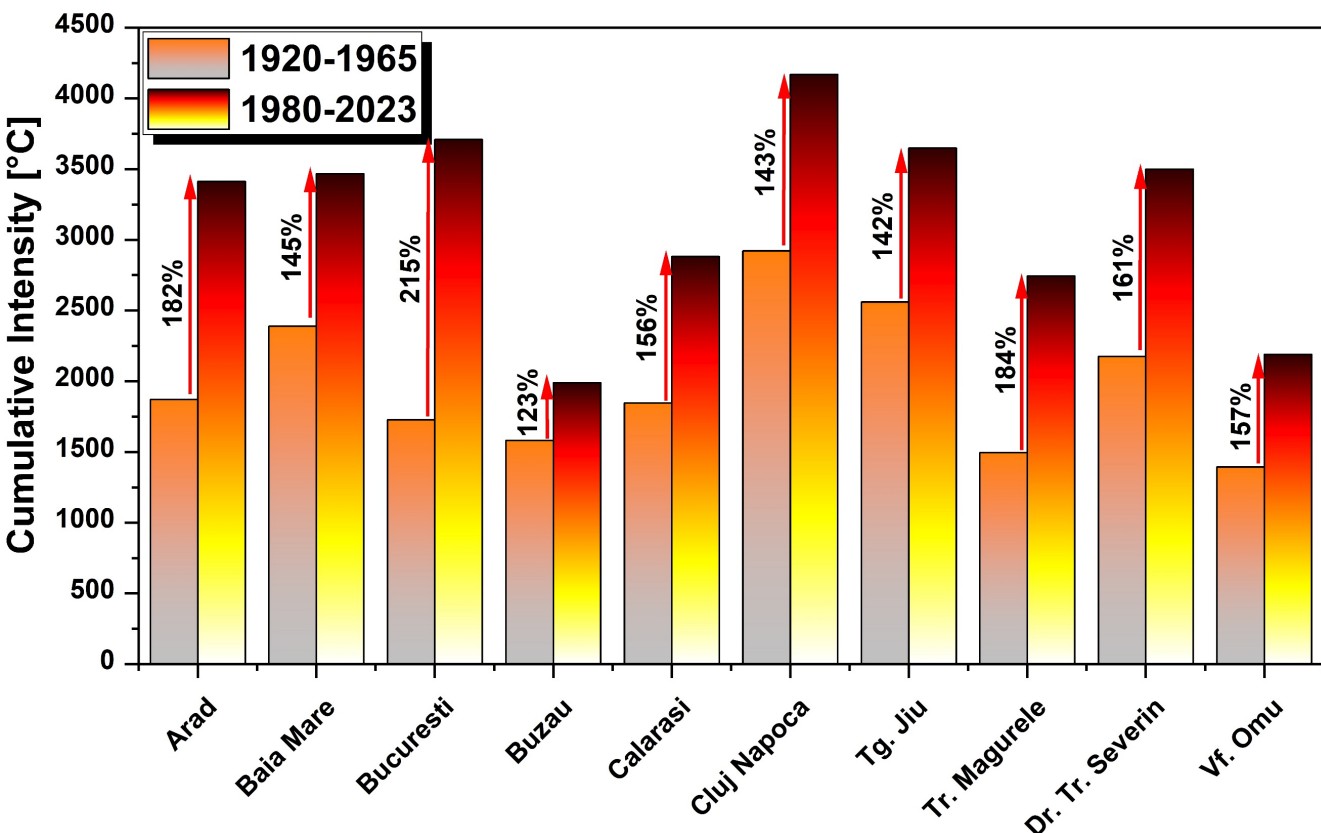

***Figure 7***. Distribution of the cumulative (i.e., sum of daily maximum temperature anomaly over all days affected by a HW) over two periods, namely 1920 – 1965 (gray-to-orange bars) and 1980 – 2023 (yellow-to-red bars), respectively. The black arrows indicate the rate of change (as %) between the two analyzed periods. Yellow (grey) colors indicate smaller values, while red (orange) colors indicate higher values.

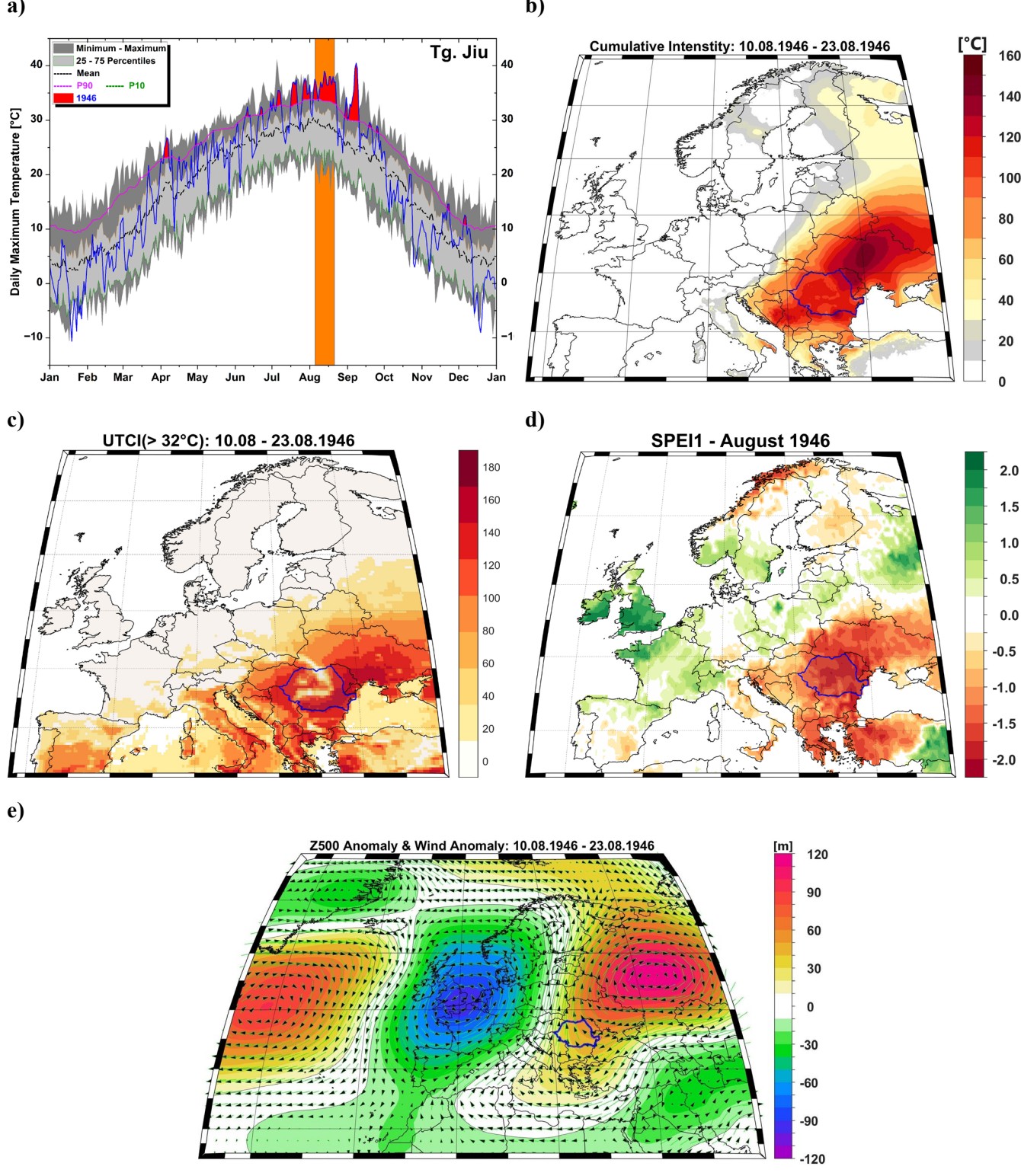

**Figure 8**. a) Daily maximum temperature at Tg. Jiu meteorological station for the year 1946; b) The cumulative intensity of the HW occurring over the period 10.08 – 23.08.1946; c) The number of hours with the Universal Thermal Climate Index>32°C throughout the duration of the HW; d) The 1-month Standardized Precipitation Evapotranspiration Index for August 1946 and e) The 500mb geopotential height anomaly and the associated winds averaged over the period 10.08 – 23.08.1946. In a) the green line represents the 10[th] percentile (P10) of the daily maximum temperature, the black dotted line represents the mean of the daily maximum temperature and the red line represents the 90[th] percentile (P90) of the daily maximum temperature. The period 1971–2000 was used to compute the daily maximum temperature climatology.

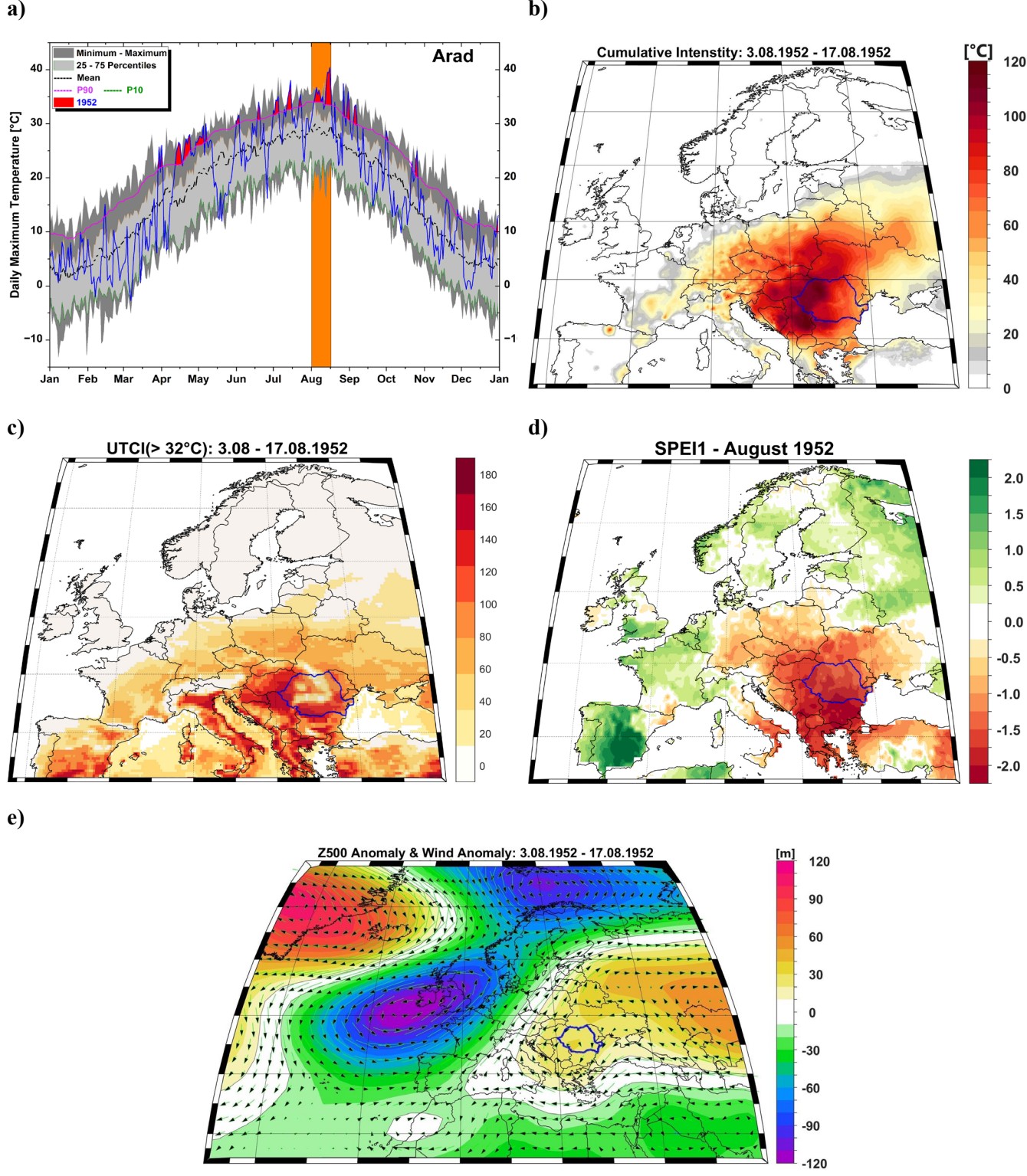

***Figure 9***. a) Daily maximum temperature at Arad meteorological station for the year 1952; b) The cumulative intensity of the HW occurring over the period 3.08 – 17.08.1952; c) The number of hours with the Universal Thermal Climate Index>32°C throughout the duration of the HW; d) The 1-month Standardized Precipitation Evapotranspiration Index for August 1952 and e) The 500mb geopotential height anomaly and the associated winds averaged over the period 3.08 – 17.08.1952. In a) the green line represents the 10th percentile (P10) of the daily maximum temperature, the black dotted line represents the mean of the daily maximum temperature and the red line represents the 90th percentile (P90) of the daily maximum temperature. The period 1971–2000 was used to compute the daily maximum temperature climatology.

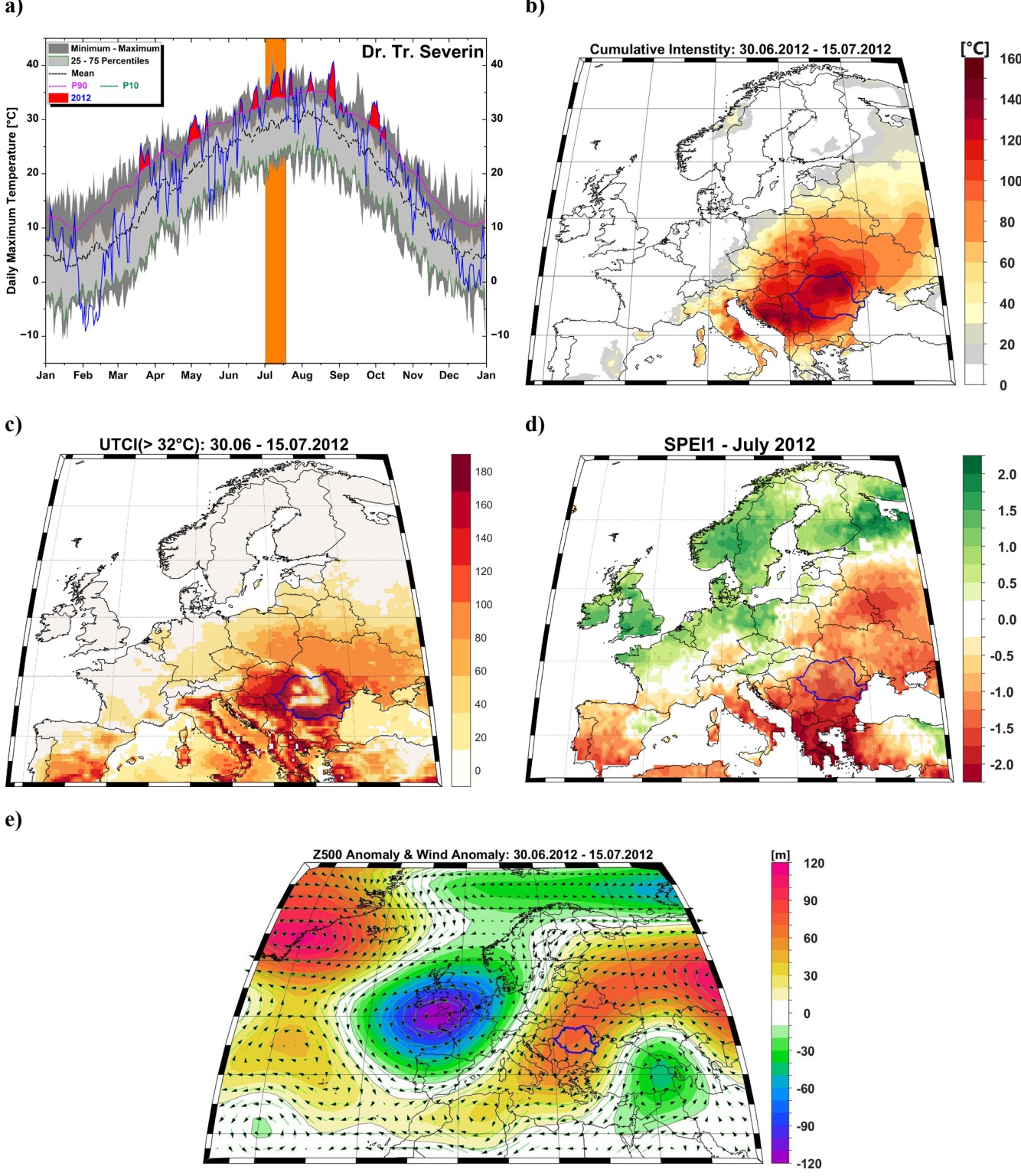

***Figure 10***. a) Daily maximum temperature at Dr. Tr. Severin meteorological station for the year 2012; b) The cumulative intensity of the HW occurring over the period 30.06 – 15.07.2012; c) The number of hours with the Universal Thermal Climate Index>32°C throughout the duration of the HW; d) The 1-month Standardized Precipitation Evapotranspiration Index for July 2012 and e) The 500mb geopotential height anomaly and the associated winds averaged over the period 30.06 – 15.07.2012. In a) the green line represents the 10[th] percentile (P10) of the daily maximum temperature, the black dotted line represents the mean of the daily maximum temperature and the red line represents the 90[th] percentile (P90) of the daily maximum temperature. The period 1971–2000 was used to compute the daily maximum temperature climatology.

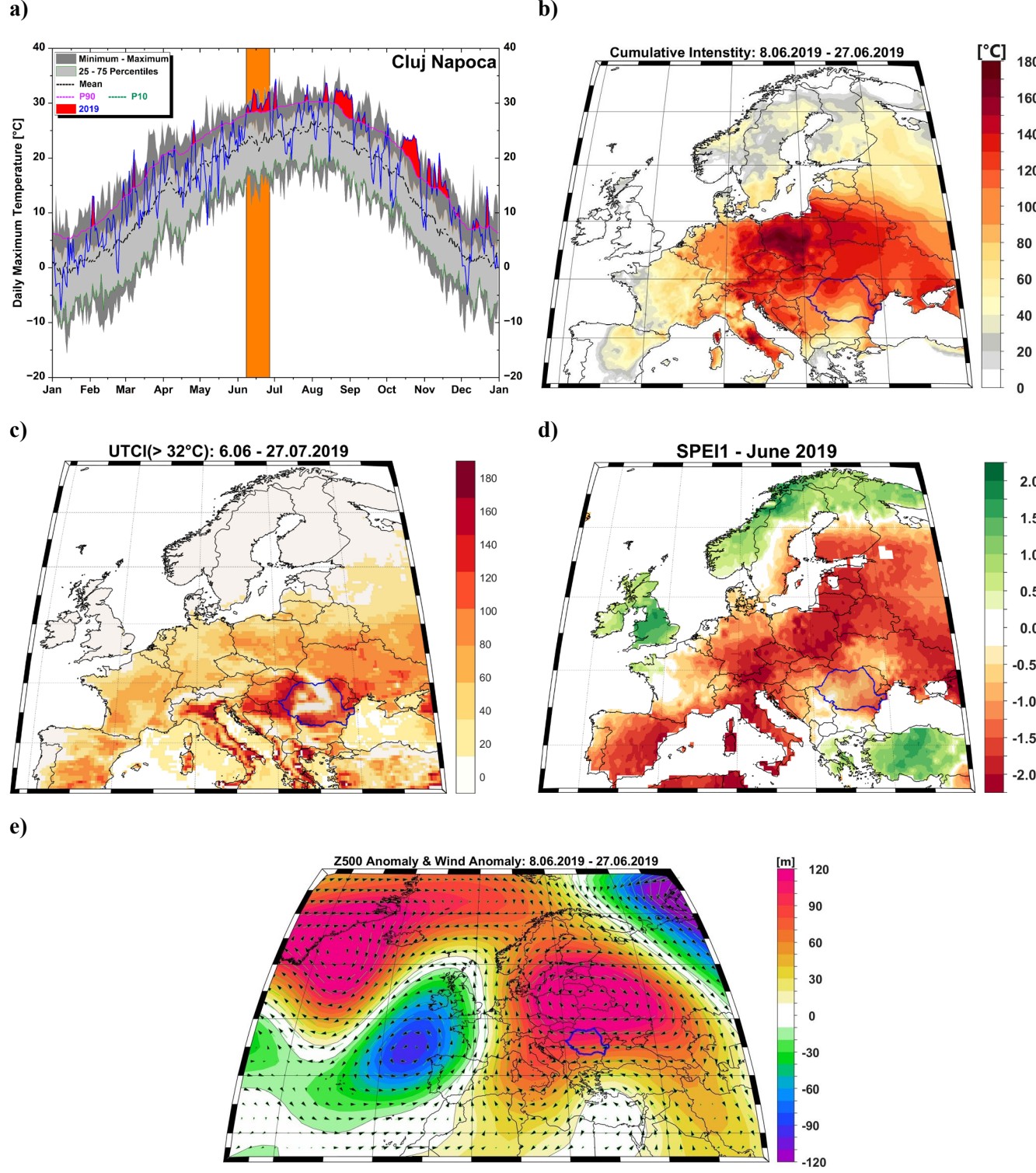

***Figure 11***. a) Daily maximum temperature at Cluj meteorological station for the year 1952; b) The cumulative intensity of the HW occurring over the period 8.06 – 27.06.2019; c) The number of hours with the Universal Thermal Climate Index>32°C throughout the duration of the HW; d) The 1-month Standardized Precipitation Evapotranspiration Index for June 2019 and e) The 500mb geopotential height anomaly and the associated winds averaged over the period 8.06 – 27.06.2019. In a) the green line represents the 10[th] percentile (P10) of the daily maximum temperature, the black dotted line represents the mean of the daily maximum temperature and the red line represents the 90[th] percentile (P90) of the daily maximum temperature. The period 1971–2000 was used to compute the daily maximum temperature climatology.

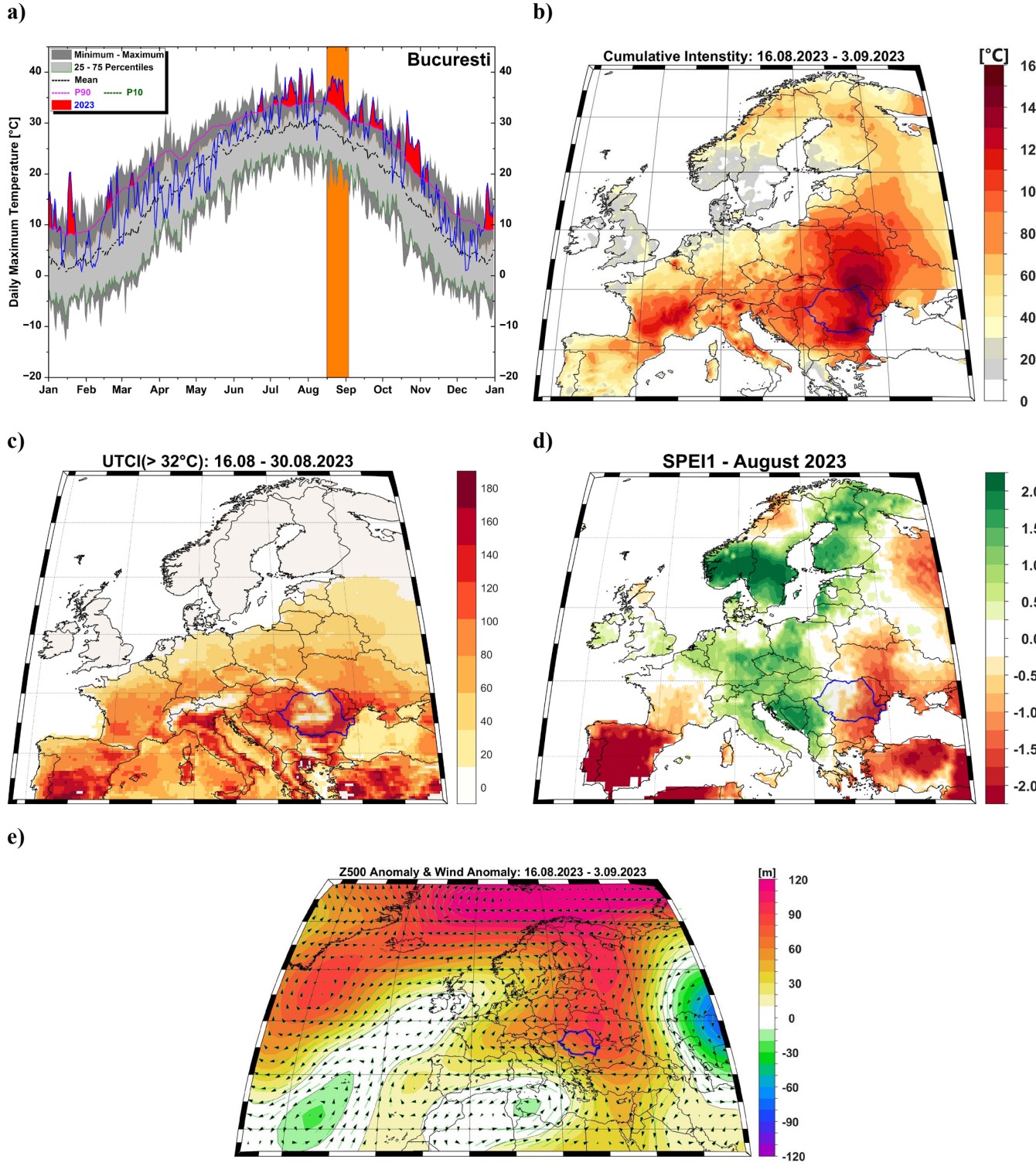

***Figure 12***. a) Daily maximum temperature at Bucuresti meteorological station for the year 2023; b) The cumulative intensity of the HW occurring over the period 16.08 – 3.09.2023; c) The number of hours with the Universal Thermal Climate Index>32°C throughout the duration of the HW; d) The 1-month Standardized Precipitation Evapotranspiration Index for August 2023 and e) The 500mb geopotential height anomaly and the associated winds averaged over the period 16.08 – 3.09.2023. In a) the green line represents the 10[th] percentile (P10) of the daily maximum temperature, the black dotted line represents the mean of the daily maximum temperature and the red line represents the 90[th] percentile (P90) of the daily maximum temperature. The period 1971–2000 was used to compute the daily maximum temperature climatology.

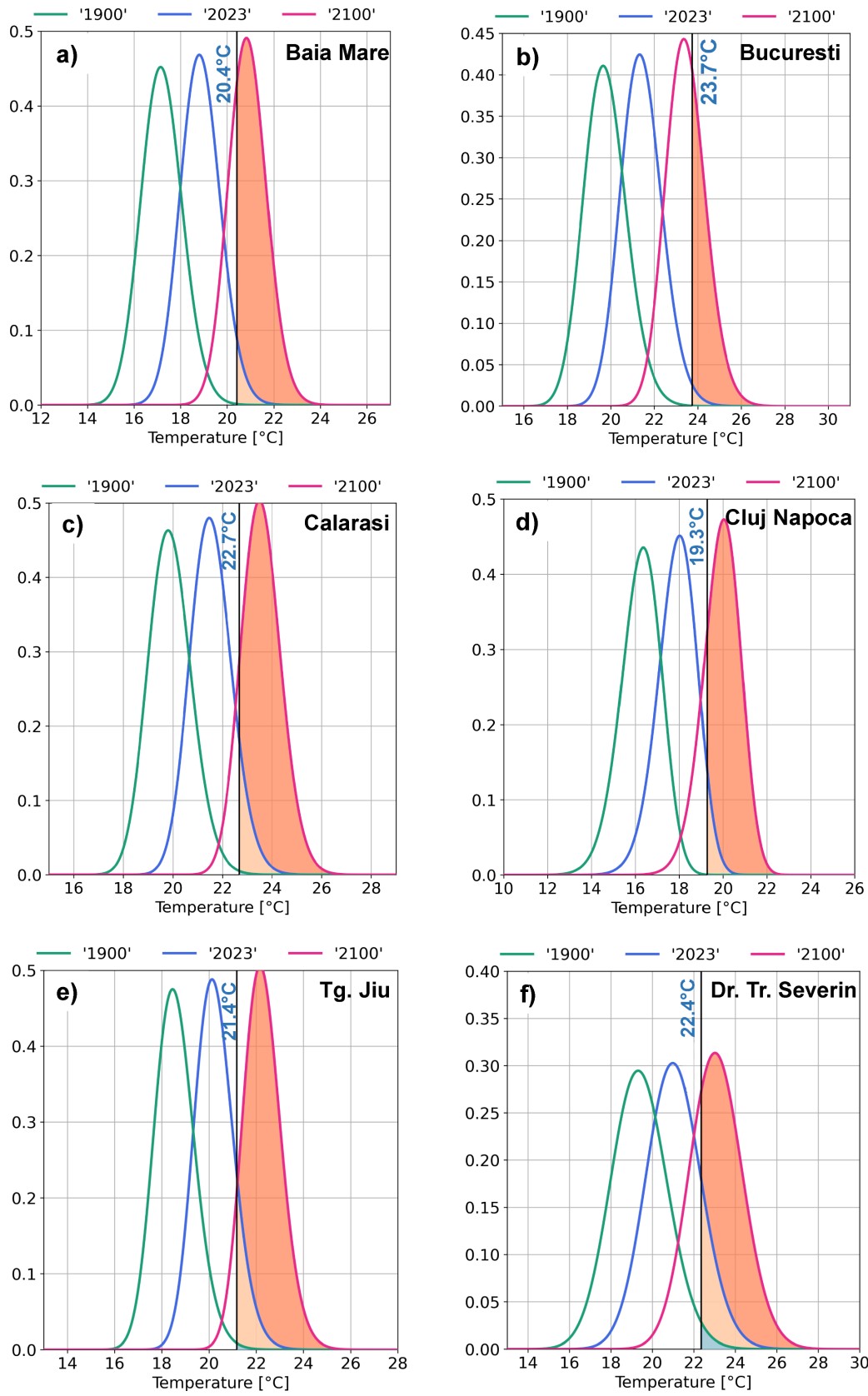

***Figure 13.*** Stochastically generated skewed (SGS) distributions of pseudo-observation for summer mean temperatures in 1900 (green line), 2023 (blue line) and 2100 (red line), for the SSP2-45 scenario. Black vertical line marks the observed summer (May-June-July-August-September) temperature in 2023. A) Baia Mare; b) Bucuresti; c) Calarasi; d) Cluj Napoca; e) Tg. Jiu and e) Dr. Tr. Severin.

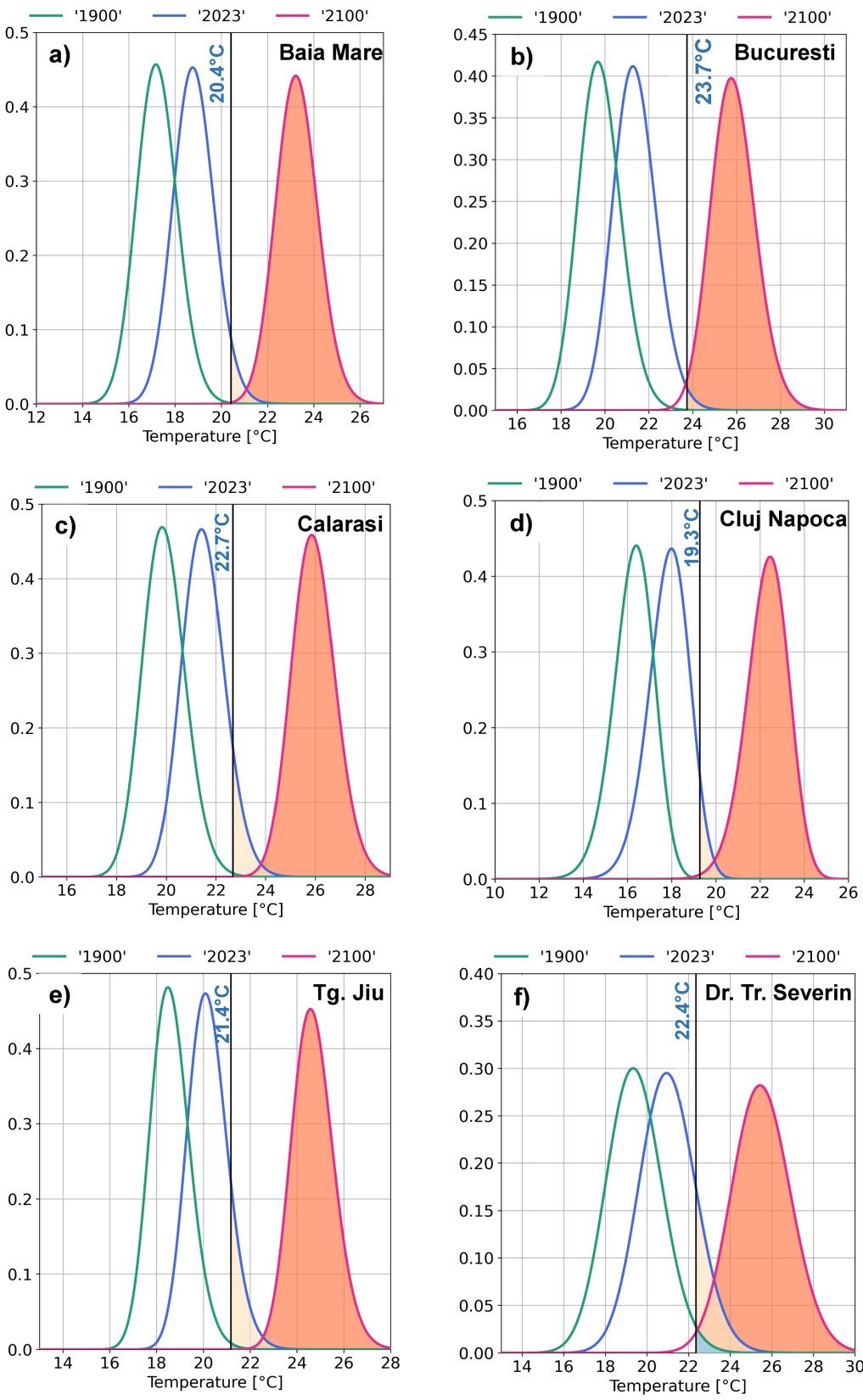

***Figure 14.*** Stochastically generated skewed (SGS) distributions of pseudo-observation for summer mean temperatures in 1900 (green line), 2023 (blue line) and 2100 (red line), for the SSP5-85 scenario. Black vertical line marks the observed summer (May-June-July-August-September) temperature in 2023. A) Baia Mare; b) Bucuresti; c) Calarasi; d) Cluj Napoca; e) Tg. Jiu and e) Dr. Tr. Severin.

**Table 1.** Names, coordinates, and length of data availability for the stations used in this study. The bolded stations are the station for which long-term observations exists. In the current study we will use the term Bucuresti for the Bucuresti Filaret station and Vf. Omul for the Varful Omu station.

| Station | Lat | Lon | Data availability (daily TX    ) |
|---|---|---|---|
| **Arad** | **46.13** | **21.35** | **1896 – 2023** |
| Bacau | 46.53 | 26.92 | 1961 - 2023 |
| **Baia Mare** | **47.67** | **23.5** | **1921 – 2023** |
| Bistrita | 47.15 | 24.5 | 1961 - 2023 |
| Botosani | 47.68 | 26.67 | 1961 - 2023 |
| Brasov | 45.65 | 25.62 | 1961 - 2023 |
| **Bucuresti Filaret (Bucuresti)** | **44.52** | **26.08** | **1885 – 2023** |
| **Buzau** | **45.13** | **26.85** | **1896 – 2023** |
| **Calarasi** | **44.21** | **27.32** | **1898 – 2023** |
| Caransebes | 45.42 | 22.25 | 1961 - 2023 |
| Ceahlau | 46.92 | 25.92 | 1961 - 2023 |
| **Cluj** | **46.78** | **23.57** | **1923 – 2023** |
| Constanta | 44.22 | 28.63 | 1961 - 2023 |
| Craiova | 44.23 | 23.87 | 1961 - 2023 |
| Deva | 45.87 | 22.9 | 1961 - 2023 |
| Galati | 45.50 | 28.02 | 1961 - 2023 |
| Iasi | 47.17 | 27.63 | 1961 - 2023 |
| M. Ciuc | 46.37 | 25.73 | 1961 - 2023 |
| Oc. Sugatag | 47.78 | 23.93 | 1961 - 2023 |
| Oradea | 47.06 | 21.93 | 1961 - 2023 |
| Rm. Valcea | 45.10 | 24.37 | 1961 - 2023 |
| Rosiorii de Vede | 44.10 | 24.98 | 1961 - 2023 |
| Satu mare | 47.79 | 22.86 | 1961 - 2023 |
| Suceava | 47.66 | 26.27 | 1961 - 2023 |
| Sibiu | 45.80 | 24.15 | 1961 - 2023 |
| Sulina | 45.17 | 29.73 | 1961 - 2023 |
| **Tg. Jiu** | **45.03** | **23.27** | **1899 – 2023** |
| **Tr. Magurele** | **43.75** | **24.88** | **1896 – 2023** |
| **Dr. Tr. Severin** | **44.63** | **22.30** | **1896 – 2023** |
| Tulcea | 45.18 | 28.82 | 1961 - 2023 |
| **Varful Omu (Vf. Omu)** | **45.45** | **25.45** | **1928 – 2023** |

*Table 2.* Eigen values of the first empirical orthogonal function (EOF1) at 9 long-term stations

| Station | Arad | Bucuresti | Baia Mare | Buzau | Calarasi | Cluj | Tg. Jiu | Tr. Magurele | Dr. Tr. Severin |
|---|---|---|---|---|---|---|---|---|---|
| Eigen Value | 0.36 | 0.36 | 0.25 | 0.33 | 0.18 | 0.32 | 0.39 | 0.37 | 0.38 |

***Table 3***. The starting and end date of the first three high-intensity HWs, based on their cumulative intensity, over the
period 1961 – 2023 recorded at 31 stations over the Romanian territory. S.D. – starting date, E.D.- ending date and
C.I. [°C]– cumulative intensity.

| Station | HW1 | | | HW2 | | | HW3 | | |
|---|---|---|---|---|---|---|---|---|---|
| | S. D. | E. D. | C.I | S. D. | E. D. | C.I | S. D. | E. D. | C.I |
| 1. Arad | 19.08.2019 | 02.09.2019 | 121.01 | 29.06.2012 | 06.07.2012 | 116.04 | 15.07.2007 | 24.07.2007 | 97.62 |
| 2. Bacau | 30.06.2012 | 09.07.2012 | 146.16 | 24.08.2015 | 06.09.2015 | 123.01 | 17.08.2023 | 29.08.2023 | 117.09 |
| 3. Bistrita | 08.06.2019 | 14.06.2019 | 139.87 | 03.09.2016 | 11.09.2016 | 128.69 | 19.08.2018 | 02.09.2019 | 121.17 |
| 4. Baia Mare | 12.06.1964 | 29.06.1964 | 114.01 | 19.08.2019 | 03.09.2019 | 223.48 | 27.07.1994 | 09.08.1994 | 109.11 |
| 5. Botosani | 14.08.2023 | 29.08.2023 | 134.55 | 25.07.2012 | 07.08.2012 | 122.32 | 02.09.2016 | 14.09.2016 | 111.39 |
| 6. Brasov | 17.08.2023 | 28.08.2023 | 111.73 | 04.09.2016 | 18.09.2016 | 100.34 | 16.07.2007 | 25.07.2007 | 96.52 |
| 7. Bucuresti | 16.08.2023 | 03.09.2023 | 154.56 | 16.07.2007 | 25.07.2007 | 95.43 | 13.06.2007 | 26.06.2007 | 91.71 |
| 8. Buzau | 17.08.2023 | 31.08.2023 | 111.73 | 20.08.2019 | 03.03.2019 | 92.52 | 16.07.2016 | 25.07.2016 | 88.60 |
| 9. Calarasi | 17.08.2023 | 30.08.2023 | 111.43 | 16.07.2007 | 25.07.2007 | 92.96 | 12.08.2008 | 24.08.2008 | 89.88 |
| 10. Caransebes | 26.06.2012 | 11.07.2012 | 108.50 | 16.07.2007 | 25.07.2007 | 95.84 | 27.08.2015 | 05.09.2015 | 95.12 |
| 11. Ceahlau | 18.08.2023 | 29.08.2023 | 98.70 | 15.07.2007 | 25.07.2007 | 96.69 | 30.06.2012 | 10.07.2012 | 96.41 |
| 12. Cluj Napoca | 08.06.2019 | 27.06.2019 | 141.18 | 29.06.2012 | 11.07.2012 | 118.70 | 17.08.2023 | 29.08.2023 | 109.39 |
| 13. Constanta | 07.06.2019 | 28.06.2019 | 137.28 | 09.09.1994 | 04.10.1994 | 126.72 | 01.08.2010 | 20.08.2010 | 108.95 |
| 14. Craiova | 19.08.2019 | 03.09.2019 | 105.19 | 17.08.2023 | 29.08.2023 | 102.11 | 15.07.2007 | 25.07.2007 | 101.78 |
| 15. Deva | 19.08.2019 | 02.09.2019 | 111.10 | 15.07.2007 | 25.07.2007 | 104.28 | 02.08.2015 | 16.08.2015 | 102.83 |
| 16. Galati | 16.08.2023 | 31.08.2023 | 120.14 | 19.08.2019 | 03.09.2019 | 111.18 | 24.08.2015 | 06.09.2015 | 110.17 |
| 17. Iasi | 14.08.2023 | 30.08.2023 | 145.13 | 24.07.2012 | 09.08.2012 | 143.07 | 24.08.2015 | 05.09.2015 | 111.39 |
| 18. Miercurea Ciuc | 01.07.2012 | 15.07.2012 | 126.02 | 17.08.2023 | 29.08.2023 | 104.96 | 03.08.2015 | 17.08.2015 | 101.37 |
| 19. Oc. Sugatag | 08.06.2019 | 27.06.2019 | 145.38 | 02.08.2015 | 16.08.2015 | 129.08 | 19.08.2019 | 03.09.2019 | 127.86 |
| 20. Oradea | 25.07.1994 | 11.08.1994 | 113.58 | 30.06.2012 | 11.07.2012 | 107.91 | 15.07.2007 | 24.07.2007 | 94.96 |
| 21. Rm. Valcea | 19.07.2012 | 09.08.2012 | 170.63 | 15.07.2015 | 30.07.2015 | 116.85 | 15.07.2007 | 25.07.2007 | 107.30 |
| 22. Rosiorii de Vede | 17.08.2023 | 29.08.2023 | 102.72 | 16.07.2007 | 25.07.2007 | 94.82 | 17.07.1987 | 27.07.1987 | 80.63 |
| 23. Satu Mare | 03.08.2015 | 16.08.2015 | 113.49 | 12.06.1964 | 29.06.1964 | 111.89 | 24.07.1994 | 09.09.1994 | 111.17 |
| 24. Sibiu | 30.06.2012 | 15.07.2012 | 129.76 | 20.08.2019 | 03.09.2019 | 106.17 | 18.08.2023 | 29.08.2023 | 91.35 |
| 25. Suceava | 02.09.2016 | 17.09.2016 | 134.29 | 25.07.2012 | 08.08.2012 | 125.26 | 03.08.2015 | 16.08.2015 | 113.96 |
| 26. Sulina | 10.09.1994 | 05.10.1994 | 122.62 | 31.07.2010 | 18.08.2010 | 116.94 | 20.08.2023 | 01.09.2023 | 75.75 |
| 27. Targu Jiu | 23.09.2012 | 20.09.2012 | 128.59 | 30.06.2012 | 16.07.2012 | 114.47 | 19.08.2019 | 03.09.2019 | 101.39 |
| 28. Turnu Magurele | 17.08.2023 | 29.08.2023 | 102.86 | 20.08.2019 | 03.03.2019 | 93.96 | 16.07.2007 | 25.07.2007 | 93.27 |
| 29. Dr. Tr. Severin | 30.06.2012 | 15.07.2012 | 116.76 | 18.08.1992 | 01.09.1992 | 107.31 | 16.07.2007 | 25.07.2007 | 93.98 |
| 30. Tulcea | 06.06.2019 | 27.06.2019 | 147.34 | 31.07.2010 | 18.08.2010 | 121.52 | 17.08.2023 | 31.08.2023 | 101.48 |
| 31. Vf. Omu | 16.07.2007 | 25.07.2007 | 88.00 | 01.07.2012 | 12.07.2007 | 82.94 | 28.08.2015 | 05.09.2015 | 71.55 |

*Table 4*. The starting and end date of the first three high-intensity HWs, based on their cumulative intensity, over the period 1885 – 2023 recorded at 11 stations over the Romanian territory. S.D. – starting date, E.D.- ending date and C.I. [°C]– cumulative intensity.

| Station | HW1 | | | HW2 | | | HW3 | | |
|---|---|---|---|---|---|---|---|---|---|
| | S. D. | E. D. | C.I | S. D. | E. D. | C.I | S. D. | E. D. | C.I |
| 1. Arad | 18.08.2019 | 02.09.2012 | 121.01 | 03.08.1952 | 17.08.1952 | 116.55 | 29.06.2012 | 11.07.2012 | 116.05 |
| 2. Baia Mare | 29.06.1932 | 16.07.1932 | 145.89 | 11.05.1937 | 22.05.1937 | 138.63 | 12.06.1964 | 29.06.1964 | 114.01 |
| 3. Bucuresti Filaret | 16.08.2023 | 03.09.2023 | 165.26 | 19.08.2019 | 03.09.2019 | 119.41 | 09.08.1946 | 23.08.1946 | 113.49 |
| 4. Buzau | 09.08.1946 | 23.08.1946 | 125.81 | 17.08.2023 | 31.08.2023 | 111.73 | 20.08.2019 | 03.09.2019 | 92.52 |
| 5. Calarasi | 10.08.1946 | 23.08.1946 | 115.55 | 17.08.2023 | 30.08.2023 | 111.43 | 16.07.2007 | 25.07.2007 | 92.96 |
| 6. Cluj Napoca | 08.06.2019 | 27.06.2019 | 141.18 | 02.08.1952 | 17.08.1952 | 134.56 | 11.05.1937 | 28.05.1937 | 129.35 |
| 7. Targu Jiu | 09.08.1946 | 23.08.1946 | 128.73 | 23.09.2012 | 07.10.2012 | 128.59 | 04.05.1907 | 20.05.1907 | 121.46 |
| 8. Turnu Magurele | 08.08.1952 | 22.08.1952 | 125.55 | 17.08.2023 | 29.08.2023 | 102.86 | 20.08.2019 | 03.03.2019 | 93.96 |
| 9. Dr. Tr. Severin | 30.06.2012 | 15.07.2012 | 116.76 | 08.08.1946 | 22.08.1946 | 114.56 | 18.08.1992 | 01.09.1992 | 107.31 |
| 10. Vf. Omu | 26.08.1956 | 07.09.1956 | 111.51 | 11.05.1958 | 21.05.1958 | 93.99 | 10.08.1946 | 22.08.1946 | 91.26 |

*Table 5.* Estimated annual probabilities, return periods, and intensities of the 2023 corresponding summer mean temperature (see the value in the brackets for each station) for the **SSP2-45 scenario**.

|  | Probability | Return period (years) | Intensity [°C] |
|---|---|---|---|
| **Baia Mare (TT =20.4°C)** | | | |
| 1900 | 0.0 (0.0 – 0.2) % | 2593 (533 – 12762) | 18.8 (18.1 – 19.3) |
| 2023 | 4.0 (1.7 – 10.2) % | 25 (10 – 58) | 20.4 |
| 2100, SSP2-45 | 72.0 (20.5 – 99.7) % | 1 (1 – 5) | 22.4 (21.6 – 23.5) |
| **Bucuresti (TT = 23.7°C)** | | | |
| 1900 | 0.0 (0.0 – 0.1) % | 5399 (984 – 142819) | 22.1 (21.4 – 22.5) |
| 2023 | 0.1 (0.0 – 0.6) % | 82 (34 – 269) | 23.7 |
| 2100, SSP2-45 | 15.6 (0.7 – 62.9) % | 3 (1 – 13) | 25.7 (24.8 – 26.8) |
| **Calarasi (TT = 22.7°C)** | | | |
| 1900 | 0.2 (0.0 – 0.6) % | 551 (154 – 4827) | 21.1 (20.6 – 21.4) |
| 2023 | 9.8 (4.8 – 18.8) % | 10 (5 – 21) | 22.7 |
| 2100, SSP2-45 | 87.9 (43.3 – 99.9) % | 1 (1 – 2) | 24.6 (23.8 – 25.5) |
| **Cluj Napoca (TT = 19.3°C)** | | | |
| 1900 | 0.0 (0.0 – 0.2) % | 15885 (552 – 355443) | 17.7 (17.0 – 18.2) |
| 2023 | 5.6 (1.8 – 15.0) % | 18 (7 – 55) | 19.3 |
| 2100, SSP2-45 | 78.9 (0.7 – 62.9) % | 1 (1 – 3) | 21.2 (20.4 – 22.3) |
| **Tg. Jiu (TT = 21.2°C)** | | | |
| 1900 | 0.3 (0.0 – 0.8) % | 295 (127 – 2163) | 19.5 (18.9 – 19.9) |
| 2023 | 13.6 (6.5 – 25.3) % | 7 (4 – 15) | 21.2 |
| 2100, SSP2-45 | 92.8 (50.3 – 100.0) % | 1 (1 – 2) | 23.2 (22.4 – 24.2) |
| **Dr. Tr. Severin (TT = 22.4°C)** | | | |
| 1900 | 1.7 (0.8 – 2.9) % | 59 (34 – 125) | 20.7 (20.1 – 21.1) |
| 2023 | 16.3 (9.5 – 25.5) % | 6 (4 – 10) | 22.4 |
| 2100, SSP2-45 | 71.5 (37.3 – 97.0) % | 1 (1 – 3) | 24.4 (23.6 – 25.4) |

*Table 6.* Estimated annual probabilities, return periods, and intensities of the 2023 corresponding summer
mean temperature (see the value in the brackets for each station) for the **SSP5-85 scenario**.

| | Probability | Return period (years) | Intensity [°C] |
|---|---|---|---|
| | **Baia Mare (TT =20.4°C)** | | |
| 1900 | 0.0 (0.0 – 0.2) % | 2727 (536 – 24486) | 18.8 (18.4 – 19.2) |
| 2023 | 4.1 (1.7 – 7.9) % | 25 (13 – 57) | 20.4 |
| 2100, SSP5-85 | 100.0 (83.6 – 100.0) % | 1 (1 – 1) | 24.9 (23.1 – 26.8) |
| | **Bucuresti (TT = 23.7°C)** | | |
| 1900 | 0.0 (0.0 – 0.1) % | 5741 (1105 – 94493) | 22.1 (21.7 – 22.4) |
| 2023 | 1.3 (0.6 – 2.9) % | 76 (34 – 180) | 23.7 |
| 2100, SSP5-85 | 98.9 (56.6 – 100.0) % | 3 (1 – 2) | 28.2 (26.5 – 30.0) |
| | **Calarasi (TT = 22.7°C)** | | |
| 1900 | 0.2 (0.0 – 0.6) % | 558 (164 – 6995) | 21.1 (20.7 – 21.4) |
| 2023 | 9.6 (4.7 – 16.6) % | 10 (6 – 21) | 22.7 |
| 2100, SSP5-85 | 100.0 (93.5 – 100.0) % | 1 (1 – 1) | 27.1 (25.4 – 28.9) |
| | **Cluj Napoca (TT = 19.3°C)** | | |
| 1900 | 0.0 (0.0 – 0.2) % | 19003 (610 – 755443) | 17.7 (17.2 – 18.1) |
| 2023 | 5.6 (2.4 – 11.9) % | 18 (8 – 42) | 19.3 |
| 2100, SSP5-85 | 99.8 (86.7 – 100.0) % | 1 (1 – 1) | 23.8 (21.9 – 25.7) |
| | **Tg. Jiu (TT = 21.2°C)** | | |
| 1900 | 0.3 (0.1 – 0.9) % | 303 (115 – 1199) | 19.6 (19.2 – 198) |
| 2023 | 13.4 (7.2 – 21.4) % | 7 (5 – 14) | 21.2 |
| 2100, SSP5-85 | 100.0 (96.1 – 100.0) % | 1 (1 – 1) | 25.7 (23.9 – 27.4) |
| | **Dr. Tr. Severin (TT = 22.4°C)** | | |
| 1900 | 1.6 (0.8 – 3.0) % | 61 (33 – 119) | 20.7 (20.4 – 21.0) |
| 2023 | 16.3 (10.6 – 23.4) % | 6 (4 – 9) | 22.4 |
| 2100, SSP5-85 | 99.0 (78.2 – 100.0) % | 1 (1 – 1) | 26.9 (25.1 – 28.5) |

834

835

836

837