# Peer review of "Examining the Eastern European heatwave of 2023 from a long-term perspective: the role of natural variability vs. anthropogenic factors"

_EGUsphere, 2024_

## Author Comment (AC1)

**Reviewer 2**

This paper examines the Eastern European heatwave of 2023 from a long-term perspective. The article selected many extreme heat wave periods in history that were accompanied by similar weather conditions and explained the high-temperature variability well through analysis. Overall, this work is very meticulous, by combining weather scale and climate scale analysis. I suggest minor revisions. Here are my detailed comments.

We thank the reviewer for the constructive evaluation of our study. In the revised version of the manuscript, we will consider all comments and suggestions and we will improve the manuscript accordingly (see detailed responses below).

1. It would help if you highlighted how to quantify the role of natural variability and anthropogenic factors in this paper, especially in the abstract and method part.

In the revised version of the manuscript, we will clarify the quantification of natural variability and anthropogenic factors, and provide a more detailed explanation of our approach and methodologies.

2. Please provide the full name of GHG

The text will be modified to include greenhouse gasses (GHGs).

3. What do the colors of the columns (bars-from yellow to black) in Figure 2 mean?

We will improve the figure(s) and add more information so they are clearer to the reader.

4. You have used many indicators to represent heat waves, but not all indicators are introduced in the methods section.

We will define/introduce all the indicators in the revised version of the manuscript.

5. I noticed that the ground stations you used in the Figures (Fig.2,4, 5, 6...) are not consistent.

Thank you for this observation. We will ensure consistency across all figures by using the same name of ground stations in Figures 2, 4, 5, and 6 for coherence and accuracy.

6. I don't know what specific ERA data you are using, please use the official data name (Thermal comfort indicators derived from ERA5 reanalysis or ERA5 hour data on pressure levels from 1940 to present) instead of ERA_ heat.

The required information will be added in the revised version of the manuscript.

7. What do the sticks in different colors below the yticks refer to?

We will improve the figure(s) and add more information so they are clearer to the reader.

8. Heatwave or HW needs to be consistent.

We will check and modify the text according to this suggestion.

9. In Fig.9, some subfigures show the study area, but some do not. Please be consistent and introduce it.

We will add the study area in all the figures.

10. I can't understand subgraph (a) in Fig8-10. The detailed information is not mentioned (i.e. green line, red area not fully displayed)

We will improve the figure(s) and add more information so they are clearer to the reader.

11. Line 155: I'm not sure if your citation method is correct, please check.

All citations will be checked and modified as suggested.

12. Line 311: please introduce the location of " Carpathian Mountains".

We will improve the figure(s) and add more information so they are clearer to the reader.

---

## Author Comment (AC2)

**Reviewer 1**

Overall, this manuscript examined the heatwaves of eastern European using long-term observation datasets. One important finding of the manuscript is to identify the importance of long-term observation data to reveal the "real" trend of heatwaves. Another important finding of the study is to reveal the correlation of heatwave and AMO. Overall, i find this paper well-written, and reads smoothly and I really enjoy reading it. Besides a few minor typos, I suggest acceptance of this manuscript. Detailed comments are listed below.

We thank the reviewer for the constructive evaluation of our study. In the revised version of the manuscript, we will consider all comments and suggestions and we will improve the manuscript accordingly (see detailed responses below).

1. In the title of the paper it mentioned "the role of natural variability vs anthropogenic factors". However, after reading the entire manuscript, i don't seem to agree with the title, as there is little about quantification of the contribution of natural and anthropogenic factors to heatwaves, such as that in Luo et al. (2023). Therefore, i would recommend either delete it, or use something else.

We agree with this point and will revise the manuscript accordingly. Specifically, we will add an analysis which deals with the attribution of the summer 2023 extreme temperature, which will add more weight to the "natural vs anthropogenic factors" discussion.

2. Line 112. should be "Which is an indicator"

3. Line 207, "which can be influenced"

4. Line 243, "indicates how difficult", i would suggest using "misleading" instead of "difficult" here

5. You may consider adding few more references here.

Lau N-C, Nath MJ (2012) A model study of heat waves over NorthAmerica: meteorological aspects and projections for the twenty-first century. J Clim 25:4761–4784. https://doi.org/10.1175/JCLI-D-11-00575.1

Yang, Z., Dominguez, F., & Zeng, X. (2019). Large and local-scale features associated with heat waves in the United States in reanalysis products and the NARCCAP model ensemble. Clim Dyn, (3), 1883–1901. https://doi.org/10.1007/s00382-018-4414-x

Chan, P.W., Catto, J.L. & Collins, M. Heatwave–blocking relation change likely dominates over decrease in blocking frequency under global warming. npj Clim Atmos Sci, **5**, 68 (2022). https://doi.org/10.1038/s41612-022-00290-2

6. Line 386 and Line 365 are the same. Please delete one of them.

All the aforementioned corrections/suggestion will be implemented in the revised version of the manuscript.

---

## Author Response (AR1)

**Reviewer 1**

Overall, this manuscript examined the heatwaves of eastern European using long-term observation datasets. One important finding of the manuscript is to identify the importance of long-term observation data to reveal the "real" trend of heatwaves. Another important finding of the study is to reveal the correlation of heatwave and AMO. Overall, I find this paper well-written, and reads smoothly and I really enjoy reading it. Besides a few minor typos, I suggest acceptance of this manuscript. Detailed comments are listed below.

We thank the reviewer for the constructive evaluation of our study. In the revised version of the manuscript, we have considered all comments and suggestions and we improved the manuscript accordingly (see detailed responses below).

1. In the title of the paper it mentioned "the role of natural variability vs anthropogenic factors". However, after reading the entire manuscript, i don't seem to agree with the title, as there is little about quantification of the contribution of natural and anthropogenic factors to heatwaves, such as that in Luo et al. (2023). Therefore, i would recommend either delete it, or use something else.

We agree with this comment, and we revised the manuscript accordingly. Specifically, we have added a new attribution analysis (detailed in Section 3.5). We also slightly changed the title of the manuscript to include the newly added analysis.

2. Line 112. should be "Which is an indicator"

Modified as suggested.

3. Line 207, "which can be influenced"

Modified as suggested.

4. Line 243, "indicates how difficult", i would suggest using "misleading" instead of "difficult" here

Modified as suggested.

5. You may consider adding few more references here.

Lau N-C, Nath MJ (2012) A model study of heat waves over North America: meteorological aspects and projections for the twenty-first century. J Clim 25:4761–4784. https://doi.org/10.1175/JCLI-D-11-00575.1

Yang, Z., Dominguez, F., & Zeng, X. (2019). Large and local-scale features associated with heat waves in the United States in reanalysis products and the NARCCAP model ensemble. Clim Dyn, (3), 1883–1901. https://doi.org/10.1007/s00382-018-4414-x

Chan, P.W., Catto, J.L. & Collins, M. Heatwave–blocking relation change likely dominates over decrease in blocking frequency under global warming. npj Clim Atmos Sci, **5**, 68 (2022). https://doi.org/10.1038/s41612-022-00290-2

The aforementioned references have been added in the revised version of the manuscript.

6. Line 386 and Line 365 are the same. Please delete one of them.

Modified as suggested.

**Reviewer 2**

This paper examines the Eastern European heatwave of 2023 from a long-term perspective. The article selected many extreme heat wave periods in history that were accompanied by similar weather conditions and explained the high-temperature variability well through analysis. Overall, this work is very meticulous, by combining weather scale and climate scale analysis. I suggest minor revisions. Here are my detailed comments.

We thank the reviewer for the constructive evaluation of our study. In the revised version of the manuscript, we considered all comments and suggestions and improved the manuscript accordingly (see detailed responses below).

1. It would help if you highlighted how to quantify the role of natural variability and anthropogenic factors in this paper, especially in the abstract and method part.

In the revised version of the manuscript, we have clarified the quantification of natural variability and anthropogenic factors, particularly in the abstract and methods sections. Additionally, we have added an attribution analysis (detailed in Section 3.5), where we employ the Rantanen et al. (2024) methodology to quantify the role of climate change in the 2023 extreme summer temperatures at several stations in Romania.

Rantanen, M., Räisänen, J., and Merikanto, J.: A method for estimating the effect of climate change on monthly mean temperatures: September 2023 and other recent record-warm months in Helsinki, Finland, Atmos. Sci. Lett., 25, e1216, 2024.

2. Please provide the full name of GHG

Corrected this. In the revised version of the manuscript, we use only the term "greenhouse gas" instead of the acronym GHG.

3. What do the colors of the columns (bars-from yellow to black) in Figure 2 mean?

We have added this clarification in the revised version of the manuscript under the caption of Figure 2: 'Yellow colors indicate smaller values while red colors indicate higher values.'"

4. You have used many indicators to represent heat waves, but not all indicators are introduced in the methods section.

We have used two heatwave indicators, namely the ones based on the $90^{th}$ percentile (TX) and the fixed threshold method (TX30). Both indicators are now introduced and explained in the data and methods section (Section 2).

5. I noticed that the ground stations you used in the Figures (Fig.2,4, 5, 6...) are not consistent.

We have included the suggestion in the revised version of the manuscript. We have used the same names in all figures/tables throughout the manuscript.

6. I don't know what specific ERA data you are using, please use the official data name (Thermal comfort indicators derived from ERA5 reanalysis or ERA5 hour data on pressure levels from 1940 to present) instead of ERA_ heat.

Modified as suggested.

7. What do the sticks in different colors below the yticks refer to?

The required information has been added in the revised version of the manuscript and in the figure caption.

8. Heatwave or HW needs to be consistent.

We have opted to consistently use 'HW' as an abbreviation for heatwave throughout the manuscript to ensure readability and avoid over-complicating the text.

9. In Fig.9, some subfigures show the study area, but some do not. Please be consistent and introduce it.

Modified as suggested.

10. I can't understand subgraph (a) in Fig8-10. The detailed information is not mentioned (i.e., green line, red area not fully displayed)

The required information has been added to the figure caption.

11. Line 155: I'm not sure if your citation method is correct, please check.

In the revised version of the manuscript, we use "(AMO, Kerr 2000)" instead of "(AMO, (Enfield et al., 2001)".

12. Line 311: please introduce the location of "Carpathian Mountains".

In the revised version of the manuscript, we have included the location of the Carpathian Mountains in Figure 1 to provide better geographic context.